# Thymol, a Monoterpenoid within Polymeric Iodophor Formulations and Their Antimicrobial Activities

**DOI:** 10.3390/ijms25094949

**Published:** 2024-05-01

**Authors:** Zehra Edis, Samir Haj Bloukh

**Affiliations:** 1Department of Pharmaceutical Sciences, College of Pharmacy and Health Science, Ajman University, Ajman P.O. Box 346, United Arab Emirates; 2Center of Medical and Bio-allied Health Sciences Research, Ajman University, Ajman P.O. Box 346, United Arab Emirates; s.bloukh@ajman.ac.ae; 3Department of Clinical Sciences, College of Pharmacy and Health Science, Ajman University, Ajman P.O. Box 346, United Arab Emirates

**Keywords:** COVID-19, antimicrobial resistance, Thymol, *Aloe vera*, smart triiodides, surgical-site infection, face mask, gauze bandage, iodophors, sustainability, sustained-release reservoir

## Abstract

Antimicrobial resistance (AMR) poses an emanating threat to humanity’s future. The effectiveness of commonly used antibiotics against microbial infections is declining at an alarming rate. As a result, morbidity and mortality rates are soaring, particularly among immunocompromised populations. Exploring alternative solutions, such as medicinal plants and iodine, shows promise in combating resistant pathogens. Such antimicrobials could effectively inhibit microbial proliferation through synergistic combinations. In our study, we prepared a formulation consisting of *Aloe barbadensis* Miller (AV), Thymol, iodine (I_2_), and polyvinylpyrrolidone (PVP). Various analytical methods including SEM/EDS, UV-vis, Raman, FTIR, and XRD were carried out to verify the purity, composition, and morphology of AV-PVP-Thymol-I_2_. We evaluated the inhibitory effects of this formulation against 10 selected reference strains using impregnated sterile discs, surgical sutures, gauze bandages, surgical face masks, and KN95 masks. The antimicrobial properties of AV-PVP-Thymol-I_2_ were assessed through disc diffusion methods against 10 reference strains in comparison with two common antibiotics. The 25-month-old formulation exhibited slightly lower inhibitory zones, indicating changes in the sustained-iodine-release reservoir. Our findings confirm AV-PVP-Thymol-I_2_ as a potent antifungal and antibacterial agent against the reference strains, demonstrating particularly strong inhibitory action on surgical sutures, cotton bandages, and face masks. These results enable the potential use of the formulation AV-PVP-Thymol-I_2_ as a promising antimicrobial agent against wound infections and as a spray-on contact-killing agent.

## 1. Introduction

Antimicrobial resistance (AMR) is one of the most dangerous issues that humankind currently faces [1,2,3,4,5]. AMR is propelled by the emergence and spread of multi-drug-resistant ESKAPE pathogens consisting of *Enterococcus faecium*, *Staphylococcus aureus*, *Klebsiella pneumoniae*, *Acinetobacter baumannii*, *Pseudomonas aeruginosa*, *Enterobacter* spp., and *Escherichia coli*. These pathogens persist in public places, hospital wards, and emergency units, causing community- and hospital-acquired nosocomial infections, respectively. However, such microorganisms contribute increasingly to higher rates of morbidity and exacerbate treatment challenges, costs, and mortality [1,2,3,4,5]. The COVID-19 pandemic intensified AMR due to the increased use of antimicrobials to treat severely ill patients with comorbidities [1,2,3,4,5,6]. Nevertheless, the COVID-19 pandemic deeply affected humanity by exacerbating feelings of insecurity through shortages of essential supplies such as medication and personal protective equipment [4,5,6,7,8]. The rapid and uncontrollable spread of the virus through airborne droplets led to a significant increase in hospitalizations and mortality rates worldwide, meeting unprepared, inadequate, and overwhelmed medical facilities [4,5,6,7,8]. Nevertheless, apart from synthetic antimicrobials, historically utilized, well-known remedies in the form of antimicrobial medicinal plants and microbicides like iodine in the form of povidone iodine could be an alternative pathway against AMR [3,5,9,10,11,12,13,14,15,16,17,18,19,20,21,22,23,24].

Historically, iodine (I_2_) has been recognized and utilized as a microbicide, albeit associated with certain adverse effects such as skin discoloration and irritation [25,26,27]. Additionally, iodine is tainted by its uncontrolled iodine release, leading to shorter activity margins during treatment regimens [25,26,27]. This problem can be solved by understanding the properties of iodine and designing a scaffold or molecular environment like PVP around iodine to prevent its uncontrolled, premature release [16,17,18,19,20,21,22,23,24,25,26,27]. 

Within various molecular structures, iodine forms diverse moieties, including molecular iodine, iodide ions, triiodide, and higher polyiodide anions like pentaiodides and heptaiodides [28,29,30,31,32,33,34,35,36,37,38,39,40,41]. Our previous investigations dealt with some of those polyiodide units within different molecular environments like 12-crown-4 or PVP, which impact their stability and antimicrobial activities [42,43,44,45,46,47,48,49,50,51,52,53]. Although triiodides [I_3_^−^] are deemed the most stable form of polyiodides, they display several different structural subtypes with different stability [25,27,28,29,30,31,32,33,34,35,36,37,38,39,40,41,42,43,45,46,49,50]. Triiodides can manifest as inherently stable, symmetrical, linear “smart” units [I-I-I^−^] and less stable, nonlinear [I-I^….^I^−^] or [I_2_^….^I^−^] configurations, contingent upon their immediate molecular environment [25,27,28,29,30,31,32,33,34,35,36,37,38,39,40,41,42,43,45,46,49,50]. The moieties, which are marked by uncontrolled iodine release, therefore diminishing long-term stability and treatment efficacy, are the nonlinear [I-I^….^I^−^] configurations [25,27,28,29,30,31,32,33,34,35,36,37,38,39,40,41,42,43,45,46,49,50]. Control over iodine release mechanisms can be achieved by promoting the formation of “smart”, linear, and symmetrical triiodides [I-I-I^−^] through precise molecular surroundings that stabilize them within compatible complexing agents [25,27,28,29,30,31,32,33,34,35,36,37,38,39,40,41,42,43,45,46,49,50]. One of the best complexing agents appears to be polyvinylpyrrolidone (PVP). Therefore, numerous products incorporate iodine into PVP as PVP-I_2_, a well-known and trusted iodophor called povidone iodine [16,17,18,19,20,21,22,23,24,25,26,27]. PVP attaches triiodide ions through hydrogen bonding and protects them from premature release. According to a study by Ma et al., PVP is termed as a sustained-release reservoir for iodine [24]. Interactions with and exposure of PVP to polar molecules, light, or oxygen are detrimental for the antimicrobial properties because triiodide ions are released [24]. PVP-I_2_ serves as a fundamental microbicide that is available in almost every pharmacy and health setting globally [16,27]. Its applications range from oral care to wound treatment and surgical procedures [16,17,18,19,20,21,22,23,24,25,26,27]. Different combinations of PVP-I_2_ with medicinal plants have been garnering interest for some time [16,17,18,19,20,21,22,23,24,25,26,27,42,43,45,46,47]. Rahma et al. studied PVP–curcumin combinations with good antimicrobial activities [47]. Such formulations could generate novel generations of natural antimicrobial agents, which have the potential to complement or substitute ineffective conventional drugs [16,17,18,19,20,21,22,23,24,25,26,27,42,43,45,46,47]. 

Throughout their existence, medicinal plants have incorporated synergistically active compounds, which are pivotal against dangerous microorganisms [9,10,11,12,13,14,15]. AMR could be mitigated by utilizing combinations of plant bioactive compounds with or without diverse nanoparticles (NPs). Our research team explored iodine-based formulations comprising silver nanoparticles (AgNPs) and/or plant extracts previously [42,43,44,45,46,47,48,49,50,51,52,53,54,55,56]. Among the materials employed in our investigations were iodophors with Aloe vera (AV) in combination with either *Salvia officinalis L.* (Sage), *Cinnamomum zeylanicum* (Cinn), or *Thymus vulgaris L.* (Thyme) [42,43,44,45,46,47,48,49,50,51,52,53,54,55,56]. 

Thyme is a globally well-known medicinal plant utilized historically against microbial infections [42,57,58,59,60,61,62,63,64,65,66,67,68,69,70]. Thyme extracts contain the antimicrobial secondary metabolites Thymol and carvacrol [3,5,9,10,11,12,13,14,42,60,61,62,63,64,65,66,67,69,70,71]. Thymol, a monoterpenoid with outstanding antimicrobial properties, appears in an increasing number of investigations in combination with other essential oils, plant extracts, nanoparticles, and chitosan [70,71,72,73,74,75,76,77,78,79,80,81,82,83,84,85,86,87,88]. Thymol is a strong antimicrobial agent, which can easily enter cell membranes due to its small size and structural properties [68,71,73,79,81,82,86]. Thymol can enter like carvacrol through the microbial cell membranes [82,86]. Thymol can pass through the pores in the Gram-negative bacterial cell membrane [68,71,73,79,81,82,86]. The polar hydroxyl group interacts through hydrogen bonding with the polar cell membrane components and compromises cell functions [68,71,73,79,81,82,86]. Its nonpolar aromatic ring engages with lipid bilayers and changes their fluidity and flexibility [82,86]. The outcome is rupture of the cell membrane, leaking cell contents, and, finally, bacterial death [68,71,73,79,81,82,86]. However, many studies are based on few microbial strains. Additionally, Thymol can be synergistically effective in combination with AV against pathogens [83]. A study from Sharma et al. studied the combination of AV and chitosan-encapsulated Thymol as an antimicrobial agent with very good inhibitory action [83]. Consequently, incorporating Thymol into PVP-I_2_ together with polyphenols or essential oils originating from plant extracts could render good antimicrobial agents. We investigated the susceptibility of 10 reference strains against Thymol alone and within the polymeric iodophor PVP-I_2_ combined with AV.

AV has been recognized across various cultures for centuries as a health-promoting, anti-inflammatory, anti-microbial, and moisturizing agent [84,85,86,87,88,89,90,91,92,93,94,95,96,97,98,99,100,101,102,103,104,105,106,107,108,109,110,111,112,113,114,115]. AV boasts a rich composition of over 75 bio-compounds, such as aloin, aloe-emodin, acemannan, emodin, galacturonic acid, and carbohydrates like mannose [42,89,90,91,92,93,94]. Moreover, AV stands out as a globally accessible, cost-effective, and sustainable resource [42,89,90,91,92,93,94]. We opted for AV gel to enhance the antimicrobial properties of our formulations based on iodine [42,43,44,45,46]. Nevertheless, we noticed surging antimicrobial activities in our latest investigation with the addition of ethanolic Thyme extract into our AV-PVP-I_2_ formulations [42]. 

Based on our previous investigations, we combined Thymol with AV, PVP, and smart triiodides in AV-PVP-Sage-I_2_ formulations through a facile, straightforward, one-pot preparation [42,43,44,45,46]. The morphology and composition were studied by UV-vis, FTIR, Raman, XRD, SEM, and EDS. Antimicrobial studies were conducted against a selection of 10 reference strains (*C. albicans* WDCM 00054 Vitroids, *S. aureus* ATCC 25923, *B. subtilis* WDCM0003, *E. faecalis* ATCC 29212, *S. pneumonia* ATCC 49619, *S. pyogenes* ATCC 19615, *K. pneumonia* WDCM00097 Vitroids, *E. coli* WDCM 00013 Vitroids, *P. aeruginosa* WDCM 00026 Vitroids, and *P. mirabilis* ATCC 29906). The formulation was impregnated on sterile discs, surgical polyglycolic acid (PGA) sutures, cotton bandages, surgical face masks, and KN95 masks in accordance with our previous investigations [42,43,44,45,46]. We also studied the antimicrobial properties of the formulation after 25 months of storage.

Our title formulation AV-PVP-Thymol-I_2_ was verified as a strong antifungal agent against *C. albicans* WDCM 00054 Vitroids. Among the most susceptible Gram-positive bacteria were *S. aureus* ATCC 25923, *B. subtilis* WDCM0003, and *E. faecalis* ATCC 29212. Moderate to intermediate inhibitory action was observed against the Gram-negative microorganisms *K. pneumonia* WDCM00097 Vitroids, *E. coli* WDCM 00013 Vitroids, and *P. aeruginosa* WDCM 00026 Vitroids depending on the impregnated material. Accordingly, the highest inhibition zones were recorded where layers of surgical and KN95 face masks, as well as gauze bandages, were used. These results indicate the potential applications of the title compound AV-PVP-Thymol-I_2_ to prevent or treat infections and act as a contact-killing agent on inanimate surfaces. Further in vivo experiments and cytotoxicity studies are needed to verify the suggested applications.

## 2. Results and Discussion

AMR is characterized by the proliferation of multi-drug-resistant ESKAPE pathogens, rendering them impervious to synthetic antimicrobials [1,2,3,4,5,6]. Such instances of treatment ineffectiveness pose a particularly grave risk among elderly and immunocompromised individuals [1,2,3,4,5,6,7,8]. New classes of synergetic bio-antimicrobials could present a solution against AMR [9,10,11,12,13,14,15,57,59,60,61]. The additional incorporation of PVP-I_2_ as a sustained-release reservoir of molecular iodine into such formulations improves their antimicrobial properties [16,17,18,19,20,21,22,23,24,25,26,27,42,43,44,45,46]. Consequently, our study delves into analyzing the morphology and composition of AV-PVP-Thymol-I_2_. Antimicrobial efficacy was assessed against a selected panel of 10 reference strains on various substrates including discs, sutures, bandages, surgical face masks, and KN95 masks.

### 2.1. Morphological Examination and Elemental Composition of AV-PVP-Thymol-I_2_

The composition and morphology of AV-PVP-Thymol-I_2_ were verified by energy-dispersive X-ray spectroscopic (EDS) and Electron Microscope (SEM) analysis, respectively (Figure 1).

Figure 1a reveals a smooth and finely homogenous morphology of AV-PVP-Thymol-I_2_ with few small, rounded structures similar to AV-PVP-Thyme-I_2_ (Figure 1a, Appendix A) [42]. The EDS in Figure 1b shows the presence of carbon (42%), oxygen (27%), iodine (3%), Cu (1.8%), and further elements originating from AV. Aluminium (24%) appears due to the sample holder [58]. Gold is used to coat the formulation, and therefore, it is seen at a low intensity in the EDS. Accordingly, the purity of the title formulation AV-PVP-Thymol-I_2_ is verified by EDS.

We coated absorbable, braided polyglycolic acid (PGA) sutures, gauze bandages, and KN95 and sterile surgical face masks with the title formulation AV-PVP-Thymol-I_2_ at a concentration of 11 µg/mL. The impregnated material was subsequently analyzed by SEM and EDS. Analyses of plain and dip-coated surgical PGA sutures are depicted in Figure 2.

Coating a plain PGA suture (Figure 2a) with the title formulation AV-PVP-Thymol-I_2_ leads to a homogenous, smooth PGA surface in the SEM (Figure 2b) [45]. Consequently, such coated sutures can alternatively be utilized to prevent surgical-site infections due to their homogenous, antimicrobial coating [116,117]. The EDS analysis depicts carbon (53%), oxygen (40%), iodine (2%), copper (4%), and chlorine (0.3%), similar to the results of AV-PVP-Thyme-I_2_ (Figure 2c, Appendix A) [42]. Copper and chlorine can be attributed to the AV components and even the suture itself, while the EDS verifies the purity of the title formulation.

During any epidemic or pandemic, personal protection equipment is pivotal for mitigating the spread of diseases and infectious agents [8,118,119,120,121,122,123,124]. Face masks prevent upper respiratory tract infections up to a certain level depending on their quality and the pathogen characteristics [8,118,119,120,121,122,123,124]. The most common ones are single-use, plain surgical face masks and KN95 masks. However, disposable face masks cause environmental pollution and are not a sustainable solution for future pandemics [124]. Dip-coating face masks with antimicrobials could increase their protective effects and enable their re-use [42,118,119,120,121,122,123]. Consequently, re-using would remove the burden on environment and public spending and presents a sustainable solution for the planet and the safety of the global population.

The SEM and EDS analysis of coated surgical facemasks confirms the purity and homogenous, smooth appearance of AV-PVP-Thymol-I_2_ (Figure 3, Appendix A).

The utilized surgical face masks comprised a white (towards the face) and a blue layer (away from face), which were separately analyzed by SEM and EDS. The white layer is a very dense network of fine, white fibres (Figure 3a). Coating the material with AV-PVP-Thymol-I_2_ results in a denser network of smoothly covered fibres (Figure 3b). The EDS of the white layer exhibits carbon (84%), oxygen (8%), iodine (4%), and copper (3%) (Figure 3c). Chlorine and potassium are almost irrelevant and are related to the face mask material or AV gel. Accordingly, the purity of AV-PVP-Thymol-I_2_ is therefore confirmed by the absence of any other unrelated element (Figure 3c). The blue layer consists of darker and thicker fibres with a nodular, uneven surface texture (Figure 3d). The EDS analysis depicts carbon (87%), oxygen (7%), copper (4%), and iodine (3%), and therefore verifies the purity of the formulation on blue face mask tissues (Appendix A). After coating the blue face mask layer, the SEM in Figure 3d presents homogenously coated, smooth fibres in contrast to the previously rough-grained fibers in Figure 3d. A detailed picture of the coating formulation AV-PVP-Thymol-I_2_ is seen in Figure 3e. The smooth surface consists of very small, almost nanoparticle-sized, smooth, round-shaped moieties like in the previous formulation, AV-PVP-Thyme-I_2_ [42]. Apparently, both formulations consist of similar semi-crystalline, small, square- and spherical-shaped compositions, which homogenously coats materials. Coating sterile surgical face masks with our homogenous, antimicrobial title formulation could mitigate airborne infections in the upper respiratory tract [42,118,119,120,121,122,123].

Gauze bandages are important tools in the prevention and treatment of wounds [98,99,100,101,102,103,125]. Suitable, antimicrobial agents dip-coated on the bandage material could prevent microbial proliferation [98,99,100,101,102,103,125]. Therefore, we impregnated AV-PVP-Thymol-I_2_ on sterile gauze bandages and studied their morphology and composition under SEM and EDS (Figure 4, Appendix A).

The title formulation AV-PVP-Thymol-I_2_ uniformly coats the cotton gauze bandage (Figure 4). In the detailed view, AV-PVP-Thyme-I_2_ and the title compound reveal the same semi-crystalline, smooth pattern of nanoscale, almost spherical, even entities like in Figure 3f [42]. EDS of the title formulation AV-PVP-Thymol-I_2_ on the cotton bandage detects carbon (54%), oxygen (38%), copper (6%), and iodine (2%). Chlorine originates from AV or the bandage, while gold is due to coating the sample with Au. In comparison, Zhan et al. coated cotton fabric with Thymol for antimicrobial purposes [76]. Their SEM also showed a homogenous, smooth pattern of coating on the cotton fabric in agreement with our results in Figure 4b.

The uniform, fine, and even coating of face masks and cotton bandages, as well as braided PGA sutures, unlocks the potential utilization of our title compound AV-PVP-Thymol-I_2_ as a contact-killing agent on those materials, as well as antimicrobial dressings.

### 2.2. Spectroscopical Characterization

#### 2.2.1. Raman Spectroscopy 

The Raman spectrum of AV-PVP-Thymol-I_2_ is presented in Figure 5.

The Raman examination of the formulation AV-PVP-Thymol-I_2_ depicted in Figure 5 validates the existence of polyiodide groups. The Raman spectrum is consistent with our earlier investigations [42,43,45,46,48,51]. The high-intensity absorption peak observed at 112 cm^−1^ signifies the presence of smart, linear, symmetric I_3_^−^ units accompanied by a minority of nonlinear, asymmetric triiodide ions. The latter moieties are verified by a broad band between 141 and 148 cm^−1^ (Figure 5, Table 1) [20,29,38,42,43,45,46,48,51,83].

The absorption peak at 112 cm^−1^ in Figure 5 is attributed to symmetrical vibrations generated by symmetrical triiodide ions [(I-I-I^−^)]. These are typically observed within the range of 100–115 cm^−1^ and were also detected in the formulation AV-PVP-Thyme-I_2_ previously (Figure 5, Table 1) [42]. Weak, broad shifts at 141 and 145 cm cm^−1^ are associated with unsymmetrical triiodide units composed of molecular iodine and I^−^ ions forming [(I-I^….^I^−^)] overtones (Figure 5, Table 1) [42,43,45,46,48,51]. These units exhibit nonlinearity, distortion, and asymmetry, as confirmed by UV spectral analysis in subsequent sections of this study (Figure 5, Table 1) [42,43,45,46,48,51]. However, the Raman spectrum reveals no indication of other iodine moieties. The absence of prominent absorption bands between 140 and 175 cm^−1^, along with the lack of absorption around 166 cm^−1^, definitively confirms the absence of I_5_^−^ ions in AV-PVP-Thymol-I_2_ like in the previous formulation AV-PVP-Thyme-I_2_ (Figure 5, Table 1) [42,43,45,46,48]. Additionally, the Raman spectrum further validates the purity of the title compound.

#### 2.2.2. UV-Vis Spectroscopy

The UV-vis spectra of fresh AV-PVP-Thymol-I_2_, 25-month-old AV-PVP-Thymol-I_2_, AV-PVP-Thymol, Thymol, and PVP-I_2_ are shown in Figure 6. 

Comparing the UV-vis spectra of fresh AV-PVP-Thymol-I_2_, 25-month-old AV-PVP-Thymol-I_2_, AV-PVP-Thymol, Thymol, and PVP-I_2_ provides an insight into the molecular processes within the formulations (Figure 6a,b). In agreement with our previous studies, there are broad absorptions between 200 and 240, 250 and 325, and 325 and 410 nm in the formulations of fresh AV-PVP-Thymol-I_2_, 25-month-old AV-PVP-Thymol-I_2_, and PVP-I_2_ (Figure 6) [42,43,45,46,48]. In particular, the iodine-related groups can be studied by UV-vis analysis, verifying the presence of iodide ions, iodine, and symmetrical and unsymmetrical triiodide units. An additional absorption in PVP-I_2_ (purple curve) is available at 444 nm, which is related to pentaiodide anions (Figure 6a) [42,43,45,46,48]. All the other iodinated samples do not contain pentaiodide moieties as already confirmed by the Raman analysis (Figure 5 and Figure 6, Table 1 and Table 2). 

However, only smart triiodide units with a linear, symmetrical structure appear red-shifted and hyperchromic at 292 nm in the title formulation (red curve) and in the 25-month-old sample compared to 290 in PVP-I_2_ (purple curve) (Figure 6b). Adding AV-Thymol into PVP-I_2_ increases the absorption intensity of triiodides at 292 nm ([I-I-I^−^]) and 360 nm ([I-I^….^I^−^]) (Figure 6b). The same happened in our previous investigation, when AV-Thyme was added to PVP-I_2_ [42]. Accordingly, the release and subsequent disintegration of pentaiodide ions (I_5_^−^) from the PVP into molecular iodine (I_2_) and triiodide anions (I_5_^−^) increases the absorption intensities of both moieties.

The absorptions of the components Thymol, PVP, AV, and iodine units overlap with each other, which consequently results in broad absorption bands [42,43,45,46,48]. Therefore, Thymol and AV bio-compounds are significantly overshadowed by the stronger-intensity absorptions of PVP and iodine moieties, posing a challenge for spectral analysis [42,43,45,46,48]. Nevertheless, comparing these spectra with pure Thymol and the previously studied AV-PVP-Thyme-I_2_ allows for better predictions regarding the composition of the title formulation (Table 2) [42]. Thymol appears clearly around 282 nm, which is red-shifted towards higher wavelengths in comparison to pure Thymol at 277 nm and 279 nm in previous reports [42,71]. This bathochromic shift indicates an increase in conjugated systems and a solvent effect leading to a reduced energy gap and less-encapsulated, free Thymol molecules within our title compound. The increased absorption intensity of the title compound compared (red curve) to pure Thymol (green curve) also confirms less-encapsulated Thymol molecules with increased availability of pi-electrons and reduced hydrogen bonding (Figure 6a).

The formulation AV-PVP-Thymol-I_2_ exhibits strong absorption bands in the UV–visible spectrum, corresponding to I_2_ at 203 nm, I^−^ at 201 nm, “smart” triiodide ions in the form of symmetrical, linear [I-I-I^−^] units at 292 nm, and nonlinear [I-I^….^I^−^] moieties at 360 nm, consistent with the formulation AV-PVP-Thyme-I_2_ and previous studies (Figure 6, Table 2) [42,43,45,46,48,51]. In comparison to AV-PVP-Thymol-I_2_, AV-PVP-Thyme-I_2_ shows similar absorbance patterns, but with less antimicrobial activity [42]. The latter compound exhibits absorption intensities for symmetrical, linear [I-I-I^−^] units at 289 nm, and nonlinear [I-I^….^I^−^] moieties at 359 nm. Accordingly, red shifts towards 292 and 360 nm in the title compound AV-PVP-Thymol-I_2_ indicate an increase in conjugated systems through the aromatic ring in Thymol in addition to the existing anthraquinones aloin, acemannan, emodin, and aloe-emodin from AV.

After storing AV-PVP-Thymol-I_2_ for 25 months (light blue curve), the UV-vis spectrum shows almost identical absorptions without any noticeable changes except higher absorption intensities (Figure 6, Table 2). This suggests that the triiodide absorption intensities, as well as their increased band broadness, indicate not only enhanced hydrogen bonding within the formulation but also a prevalence of symmetrical [I-I-I^−^] over asymmetrical triiodide [I-I^….^I^−^] moieties. This assumption is further supported by the previous Raman spectrum, revealing a strong absorption peak at 112 cm^−1^ for “smart” triiodides and a broad, weak absorption band at 141 and 145 cm^−1^ for asymmetric [I-I^….^I^−^] units (Figure 5, Table 1) [42,43,45,46,48,51]. Additionally, there are remarkable increases in absorption intensities and bathochromic red shifts related to Thymol and aloin in the 25-month-old sample. These consist of shifts for aloin from 206 to 209 nm. Thymol shows red shifts from 203 to 205 nm, 209 to 212 nm, 218 to 220 nm, and 222 to 223 nm (Figure 6, Table 2). Such red shifts towards the longer-wavelength region suggest a release of Thymol from the polymeric PVP complexation in exchange of triiodide moieties. Interestingly, there is a sharp reduction in iodine absorption at 203 nm in the 25-month-old sample. This observation indicates the loss of active iodine molecules from the PVP during storage, although the colour of the sample did not noticeably change during the 25 months. Consequently, reduced inhibitory action against the tested pathogens is expected. These assumptions are verified in the antimicrobial testing results depicted below in the coming sections of this study.

Initially, Thymol and the bioactive constituents of AV could potentially compete with triiodides for complexation by PVP via hydrogen bonding. Certain plant components from AV may disrupt this process by forming hydrogen bonds in place of triiodide ions, consequently facilitating their release from PVP. Such interference could compromise the antimicrobial efficacy of the formulation under investigation. Notably, the antimicrobial test results of the fresh title compound exhibit remarkable efficacy. 

However, lower inhibitory activity is observed after 25 months of storage in comparison to the long-term study of the formulation AV-PVP-Thyme-I_2_ [42]. The latter exhibited even higher inhibition zones after 18 months compared to the initial fresh formulation AV-PVP-Thyme-I_2_ [42]. As a result, the plant-based formulation AV-PVP-Thyme-I_2_ derived from an ethanolic Thyme extract has increased antimicrobial activity due to the synergy of the spectrum of bio-compounds available naturally within the macerated sample [42]. Those bio-compounds stabilize the polymeric iodophor and prolong, and even enhance, the sustained-release reservoir. This happens through protecting the iodine molecules captured within the PVP, inhibiting their release and the decomposition of the formulation AV-PVP-Thyme-I_2_ [42]. This synergetic, protective, and sustaining effect is missing in the title formulation AV-PVP-Thymol-I_2_ because the natural biocomponents of Thyme are missing [42]. Therefore, the sustained-release reservoir is not protected and starts to slowly release the active species iodine over a period of 25 months. Consequently, this results in reduced antimicrobial action after 25 months of storage before being exposed to any microorganism. However, the antimicrobial testing on the 10 pathogens still shows good to intermediate results.

In summary, the presence of Thymol and AV bio-compounds in the AV-PVP-Thymol-I_2_ formulation does not diminish its inhibitory effectiveness in the fresh sample. After prolonged storage of 25 months, there is reduced antimicrobial action. PVP effectively shields triiodide ions within the formulation through hydrogen bonding, serving as a sustained-release reservoir by impeding the decomposition of triiodide ions. In contrast, without this protective mechanism, iodine release would occur, leading to the discoloration of AV-PVP-Thymol-I_2_ and diminished antimicrobial efficacy over time. However, our findings from UV spectral analysis and antimicrobial testing confirm the absence of discoloration, with a slight reduction in iodine and iodide ion concentrations and enhanced antimicrobial properties observed after 25 months.

In conclusion, both AV biocomponents and Thymol demonstrate an inability to compete with triiodide ions by hydrogen bonding and encapsulation through PVP due to their bigger size. Once iodine is added to AV-PVP-Thymol, AV biocomponents and Thymol are replaced by the smaller triiodide ions. Even after a 25-month period within the formulation, triiodide ions remain shielded by PVP without being released. The majority of those triiodide anions are indeed (according to the UV absorption intensities at 292 nm) smart, symmetrical, linear [I-I-I^−^] units, which was also confirmed by Raman data. Analytical findings indicate no alteration in composition, while antimicrobial testing reveals similar inhibition between the 25-month-old sample and the fresh AV-PVP-Thymol-I_2_ formulation. Consequently, prolonged storage results in an increased availability of triiodide ions encapsulated by PVP within a sustained-release reservoir [42,43,45,46]. The absence of pentaiodide ions in the title biohybrid is not only confirmed by the previously discussed Raman spectrum but also by UV spectral analysis, and also happens within the previously investigated compound AV-PVP-Thyme-I_2_ [42]. The presence of a broad band around 444 nm in the purple curve for PVP-I_2_ indicates the presence of I_5_^−^ units, which is lacking in AV-PVP-Thymol-I_2_ (red curve) and AV-PVP-Thymol-I_2_ (after 25 months, light blue curve) (Figure 6, Table 2). 

Similar results were also found in the previously investigated AV-PVP-Thyme-I_2_ [42]. However, the results in the title compound are more pronounced because it contains pure Thymol, while the Thyme extract is composed of many constituents.

#### 2.2.3. Fourier-Transform Infrared (FTIR) Spectroscopy

The FTIR spectra of AV-PVP-Thymol and AV-PVP-Thyme-I_2_ confirm the purity and similarity of the formulations (Figure 7).

The FTIR spectra of AV-PVP-Thymol and AV-PVP-Thymol-I_2_ are almost congruent and in agreement with our previous work (Figure 7) [42,43,45,46]. AV-PVP-Thymol-I_2_ and its 25-month-old formulation show almost the same absorbance pattern and intensity. In AV-PVP-Thymol-I_2_, the bands between 3700 and 3000 cm^−1^ display an increased absorbance compared to AV-PVP-Thymol and the 25-month-old formulation (Figure 7). The enhanced absorption intensity of the available -COOH and -COH functional groups in AV-PVP-Thymol-I_2_ points to a decrease in hydrogen bonding after adding iodine to AV constituents and Thymol. The hydroxyl and carboxyl groups absorb more because they are less encapsulated by hydrogen bonding with the PVP carbonyl groups. Consequently, adding iodine into the formulation means a replacement of those compounds by triiodide ions on the PVP backbone. However, after 25 months of storage, the hydrogen bonding increased again, indicating iodine release and replacement with AV or Thymol biocomponents (Figure 7). The same increase in intensities after adding iodine happened in the region between 1500 and 1000 cm^−1^ for the in-plane bending and twisting vibrations of the methyl and methylene groups (AV, Thymol), as well as the stretching vibrations of the C-O and C-N groups (PVP, Thymol, AV) (Figure 7, Table 3). 

Meanwhile, a mixed picture is portrayed in the region between 3000 and 2800 cm^−1^ with higher absorption intensities for the title formulation at 2990 and 2855 cm^−1^ related to symmetric and asymmetric C-H stretching vibrations, respectively (Figure 7, Table 2) [42,43,45,46]. These bands show higher absorbance after adding iodine and therefore belong to the released Thymol and the AV components. Iodine competes with the latter for the PVP carbonyl groups, forms triiodide ions, and replaces them. After 25 months, the intensities decrease again. This finding verifies a release of iodine, the formation of carbonyl compounds on the PVP, and, subsequently, the binding of Thymol and AV components by hydrogen bonding (Figure 7, Table 3 (A and C)). The bands around 2949 cm^−1^ show lower absorption intensities for the iodinated title compound. These bands can be attributed to the C-H stretching vibrations originating from the PVP backbone. The exchange of Thymol and AV components with smart triiodides resulted in an additional red shift for the band at 2953 to 2949 cm^−1^, verifying the coiling of the PVP backbone (Table 3) [42,43,45,46,47]. Hydrogen bonding with triiodide molecules leads to higher encapsulation of the C-H bonds in PVP due to the complexation. This is reversed after storing the sample for 25 months, with a clear blue shift back from 2949 to 2953 cm^−1^ (Figure 7, Table 3). The bands around 1759 to 1665 cm^−1^ are very weak in both samples and indicate a very low acetylation degree with almost an absence of asymmetric -C=O stretching vibrations derived from PVP-I_2_. It is noteworthy that the absorption frequency changed slightly with a blue shift from 1757 to 1759 cm^−1^ after adding iodine, while the intensity reduced slightly. These results verify a seamless exchange of hydrogen-bonded Thymol and AV biocomponents with triiodide ions on the carbonyl -C=O of the polymeric PVP backbone. Again, after storing the sample for 25 months, a red shift occurs with higher absorption intensities from 1759 to 1757 cm^−1^ and a blue shift from 1759 to 1761 cm^−1^, reversing the settings and confirming iodine release (Table 3). Another interesting fact is the highest intensity at 1757 and 1661 cm^−1^ for the 25-month-old sample compared to the other formulations (Figure 7, inlet). This finding confirms an increase in the acetylation degree of asymmetric -C=O stretching vibrations from PVP-I_2_ by partial release of triiodide ions. In comparison to our previous investigation with AV-PVP-Thyme-I_2_, storing the sample did not lead to increasing the acetylation degree, nor was there any substantial change in the FTIR analysis. Consequently, Thymol alone is not as effective at retaining the stability of the sustained-release reservoir of the formulation as the Thyme extract. The synergistic bio-compounds within the Thyme extract stabilize the formulation even after prolonged storage and do not allow premature release of iodine, hence improving the antimicrobial properties of the AV-PVP-Thyme-I_2_ over time [42]. As a result, the natural plant extracts from Thyme prevent premature release of iodine by stabilizing the sustained-release reservoir through the synergistic action of biocomponents [42]. In comparison, Thymol in AV-PVP-Thymol-I_2_ has higher antimicrobial activities but cannot sustain the stability of the formulation over time due to the lack of other supporting, synergistic biocomponents.

However, further carbonyl groups related to AV biocomponents like aloin, acemannan, aloe-emodin, emodin, mannose, salicylic acid, uric acid, cinnamic acid, glucomannan, galacturonic acid, and trans-rosmarinic acid seem to engage into hydrogen bonding with each other and/or are reduced to -C-O^−^ or hydroxyl groups (Figure 7, Table 3) [42,43,45,46,62,67,92,93,97]. Nevertheless, the higher absorption intensities in the region 3700 and 3000 cm^−1^ belonging to the AV components and Thymol prove an increase in the number of free carboxylic -COO and –C(=O)OCH_3_ ester groups after adding iodine (Figure 7, Table 3) [42,43,45,46,62,67,92,93,97]. This assumption is based on the increased absorption intensity of the bands at 3480, 3464, 3425, 3362, 3235, 3169, and 3152 cm^−1^ after adding iodine and agrees with our previous studies (Figure 7, Table 3) [42,43,45,46,62,67,92,93,97].

In conclusion, adding iodine into AV-PVP-Thymol exchanges all hydrogen-bonded Thymol and AV biocomponents with the smaller triiodide ions with verified increased absorption intensities along with broadening due to their -C=C-, -C=O-, and -COOH- groups in the formulation. As a result of strong hydrogen bonding between the smart triiodide ions and the PVP polymetric matrix, the PVP backbone is coiled and more entangled. This is indicated in the reduced and red-shifted absorptions of the stretching vibrations related to -C-H and methylene groups available in the PVP in the form of [–CH-CH_2_-]_n_- groups. The polymeric PVP complexes triiodide anions strongly by bending around the stable smart triiodide groups and protects them by forming a sustained-release reservoir [24,42,43,45,46]. The coiled or entangled structure is confirmed in the SEM images of the bandages and the face mask in the form of small, circular or cube-like patches covering the surface of the tissues (Figure 3d,b) [42]. Storing AV-PVP-Thymol-I_2_ for 25 months slightly reverts the previous uptake of iodine by partial exchange with Thymol, which is confirmed by a higher acetylation degree in the older formulation.

### 2.3. X-ray Diffraction (XRD)

AV-PVP-Thymol-I_2_ is, according to the XRD analysis, a pure formulation composed of AV, Thymol, PVP, and iodine with a high level of crystallinity (Figure 8). 

The comparison of AV-PVP-Thymol-I_2_ and AV-PVP-Thymol shows almost the same pattern in the XRD analysis (Figure 8). AV-PVP-Thymol-I_2_ (red) depicts higher intensities than AV-PVP-Thymol (blue) (Figure 8). Adding iodine into the formulation increases the intensities of the peaks related to iodine at 2θ *=* 24, 30, and 45° in accordance with the results of previous investigations (Figure 8) [35,40,42]. PVP-related peaks increase at 2θ 14.89 and 24.37° (Table 4, Figure 8) [42,45,46]. 

Previous investigations assign 2Theta values around 12 to 25° to Thymol [42,83]. Our XRD analysis revealed similar 2Theta values at 24.43, 28.83, 30.08, and 30.14° (Table 4) [45,46,83]. After adding iodine, the peak at 28.83° vanishes and the peak at 30.08° intensifies in accordance. In comparison with AV-PVP-Thyme-I_2_, the peaks at 2Theta 14.92 and 14.98° are not available in our title compound with Thymol [42]. No further peaks are detected other than the available ones after adding molecular iodine to AV-PVP-Thymol (Figure 8, Table 4). This verifies the purity of the samples as well as the amorphization of iodine, as witnessed in our previous investigations [42,45].

The AV bio-compounds can be assigned to the peaks around 45.8, 46, 50, 62.5, and 62.7° (Table 4) in accordance with previous studies [42,45,46,83].

### 2.4. Antimicrobial Activities of AV-PVP-Thymol-I_2_

A disc diffusion assay (DD) was used to test the title formulation against 10 reference strains (*C. albicans* WDCM 00054 Vitroids, Gram-positive bacteria *S. pneumonia* ATCC 49619, *S. aureus* ATCC 25923, *S. pyogenes* ATCC 19615, *E. faecalis* ATCC 29212, and *B. subtilis* WDCM0003, as well as the Gram-negative *E. coli* WDCM 00013 Vitroids, *P. mirabilis* ATCC 29906, *P. aeruginosa* WDCM 00026 Vitroids, and *K. pneumonia* WDCM00097 Vitroids) on sterile discs with concentrations of 11, 5.5, and 2.75 µg/mL. Additionally, polyglycolic acid (PGA) sutures, cotton bandages and KN95 and surgical face masks were coated with AV-PVP-Thymol-I_2_ at a concentration of 11 µg/mL against the same 10 reference pathogens. Ethanol and water were used as negative controls and showed no inhibitory action. All the results were compared to the general antibiotics gentamycin (G) and nystatin (NY) (Table 5).

Table 5 represents the susceptibility of microorganisms towards AV-PVP-Thymol-I_2_ in decreasing order from *C. albicans* WDCM 00054 at the top to *P. mirabilis* ATCC 29906 at the end of the list (Table 5). The results verify AV-PVP-Thymol-I_2_ as a strong antifungal agent against *C. albicans* WDCM 00054 on discs (ZOI = 61, 50, 40 and 19 mm), sutures (15 mm), cotton bandages (30 mm), face masks (66 and 84 mm), and KN96 masks (+80 mm) (Table 5). The bioformulation AV-PVP-Thymol-I_2_ shows generally higher antimicrobial action against the utilized Gram-positive pathogens, followed by Gram-negative strains (Table 5). *P. mirabilis* ATCC 29906 is inherently resistant to our title formulation AV-PVP-Thymol-I_2_ in all cases and with all materials (Table 5). 

The preparation of AV-PVP-Thymol-I_2_ in methanol as a solvent renders an inhibitory zone of 72 mm in DD against *C. albicans* WDCM 00054 compared to ethanol with 61 mm (Table 6). 

Meanwhile, all the other reference strains are less susceptible to the methanolic formulation (Table 5). Increased antifungal activities are achieved by utilizing methanol as a solvent. However, the other nine microorganisms show lower susceptibilities (Table 5). Therefore, formulations of AV-PVP-Thymol-I_2_ with ethanol as the solvent present the better, non-toxic choice. 

We tested the control formulation PVP-I_2_ to judge the antimicrobial effectivity of our title compound AV-PVP-Thymol-I_2_ (Table 6). All 10 pathogens are more susceptible to the title compound with only one resistance against *P. mirabilis* ATCC 29906 (Table 6). In comparison, two strains, *P. mirabilis* ATCC 29906 and *P. aeruginosa* WDCM 00026, are resistant to PVP-I_2_ (Table 6). The inhibitory action of the previously reported AV-PVP-I_2_ is much lower in comparison to AV-PVP-Thymol-I_2_ against the same selection of pathogens [46]. Adding AV enhances the antimicrobial properties of the formulation. These findings confirm the superiority of the title compound compared to the control PVP-I_2_ and AV-PVP-I_2_ (Table 6). As a result, adding Thymol and AV increased the susceptibility of the microorganisms against the title formulation (Table 6).

Another interesting finding is the high inhibitory action of pure Thymol dissolved in ethanol at a concentration of 100 µg/mL (Table 6). In disc diffusion studies, Thymol alone shows high antimicrobial activity against Gram-negative bacteria (ZOI = 39 and 34 mm), while AV-PVP-Thymol-I_2_ (ZOI = 17 and 19 mm) shows activity towards *K. pneumoniae* WDCM 00097 and *E. coli* WDCM 00013, respectively (Table 6). *P. aeruginosa* WDCM 00026 is resistant against ethanolic Thymol solutions, but is inhibited by with a ZOI of 15 mm (Table 6). The lower Thymol concentration of the title formulation results in reduced susceptibility of Gram-negative bacteria, while Gram-positive bacteria do not differ much (Table 6). Nevertheless, improved antifungal properties are evident against *C. albicans* WDCM 00054 for pure Thymol, with a ZOI of 45 mm compared to 60 mm in AV-PVP-Thymol-I_2_ (Table 6). The title formulation presents a better tool with lower Thymol concentrations in synergy with AV biocomponents within the polymeric iodophor as a sustained-release reservoir for molecular iodine. Zhou et al. studied pure Thymol at a concentration of 100 µg/mL against S. aureus strains and reported that a slight resistance phenomenon occurs after 30 generation passages with Thymol [78]. Thymol showed fatal activity at a concentration of 100 µg/mL against several strains of the multi-drug-resistant pathogen *S. aureus* by increasing cell membrane permeability through depleting NADPH [78]. The authors also indicated that a concentration of 400 µg/mL did not result in resistance against Thymol. They suggested that these results are due to the higher concentration of 400 µg/mL and the multifaceted targets of Thymol within the pathogen [78]. A lower concentration of Thymol (100 µg/mL) within the synergetic formulation of a plant-based iodophor AV-PVP-Thymol-I_2_ could present a better strategy to combat bacterial resistance.

The formulation was kept for 25 months and again underwent antimicrobial testing. The formulation AV-PVP-Thymol-I_2_ achieved a lower ZOI and therefore showcased lower inhibitory action against the same 10 microorganisms after 25 months of storage in comparison to the fresh sample (Table 5). All ZOIs reduced by almost half, although the UV-vis analysis showed higher absorbance in almost all regions compared to the fresh sample (Figure 6b).

Compared to the plant-based AV-PVP-Thyme-I_2_ formulation, AV-PVP-Thymol-I_2_ shows less inhibitory action after 25 months of storage (Table 6) [42]. These results originate from missing synergetic bio-compounds, which enrich the natural plant extract of Thyme, in our previous study [42]. The synergistic action of the Thyme-based bio-compounds prevent the premature release of iodine. Therefore, they enhance the encapsulation of triiodide ions in the polymeric PVP complex, prevent decomposition, and mitigate the sustained-release reservoir mechanism of AV-PVP-Thyme-I_2_ [24,42]. 

In conclusion, incorporating pure Thymol into formulations highly enhances the antimicrobial action of the formulation AV-PVP-Thymol-I_2_ (Table 5). However, this effect diminishes slowly after storing the formulation for two years due to the lack of other synergetic natural bio-compounds, which assist the complexation of triiodides. During storage, Thymol seems to compete over time with smart triiodides for the hydrogen-bonded positions on the carbonyl oxygen of the PVP. This suggestion is undermined by the strong hydrogen bonding capability of Thymol [79].

The disc diffusion (DD) tests were performed with concentrations of 11, 5.5, 2.75, and 1.38 µg/mL (Figure 9, Table 5).

AV-PVP-Thymol-I_2_ is a strong antifungal agent on discs at concentrations of 11, 5.5, and 2.75 µg/mL against *C. albicans* WDCM 00054, even after storing for 25 months and with methanol (72 mm) (Figure 9, Table 5 and Table 6). Among the Gram-positive pathogens, *S. aureus* ATCC 25932 is the most susceptible to the title compound with 35, 30, and 25 mm ZOIs (Figure 9b, Table 5 and Table 6). The Gram-negative microorganism *E. coli* WDCM 00013 is the most inhibited amongst its peers with 19, 16, and 13 mm ZOIs (Figure 9c, Table 5 and Table 6). In comparison, similar patterns of susceptibility exist in the previously studied formulation AV-PVP-Thyme-I_2_ (Table 6) [42]. Both formulations, AV-PVP-Thymol-I_2_ and AV-PVP-Thyme-I_2_, are strong antifungals on discs, followed by Gram-positive and finally Gram-negative reference strains (Table 6). This pattern of inhibitory action is due to the structural outer-cell-membrane morphology of the related pathogens. *C. albicans* and Gram-positive pathogens are more susceptible to our formulations due to their less negatively charged, lipophilic outer membranes. Gram-negative pathogens have more negatively charged outer cell membranes and are more hydrophilic. Their porin channels allow passage to only negatively charged, small hydrophilic ions like iodide and triiodide ions. Iodine molecules are lipophilic and pass easily through the lipophilic peptidoglycan layers present in Gram-positive pathogens. Another discouraging factor is the motility of Gram-negative microorganisms. Indeed, the higher the motility (from top to bottom in Table 5 and Table 6), the less susceptible the bacteria are towards our formulations. Motility is inversely related to the susceptibility towards our polymeric iodophors. Consequently, both formulations face resistance by the swarming bacteria *P. mirabilis* ATCC 29906 (Table 5 and Table 6).

As a result, discs impregnated with AV-PVP-Thymol-I_2_ show strong antifungal activities against *C. albicans* WDCM 00054, followed by the Gram-positive bacteria *S. aureus* ATCC 25923, *B. subtilis* WDCM 00003, and *S. pyogenes* ATCC 19615. Good results were recorded towards *E. faecalis* ATCC 29212 and *S. pneumoniae* ATCC 49619 as well. Intermediate susceptibility towards the Gram-negative bacteria *K. pneumoniae* WDCM 00097 (ZOI = 17), *E. coli* WDCM 00013 (ZOI = 19), and *P. aeruginosa* WDCM 00026 (ZOI = 15) compared to the antibiotic Gentamycin is promising (Table 5). All the results in Table 5 verify the potential use of AV-PVP-Thymol-I_2_ as a surface contact-killing agent and antibacterial coating material. 

Surgical sutures play a pivotal role in closing open wounds during surgical procedures [42,43,44,45,116,117]. Particularly, PGA sutures are widely utilized in various medical contexts, including oral surgery [42,43,44,45,116,117]. Resistant, opportunistic microorganisms or conditions conducive to microbial proliferation can result in surgical-site infections, which can impede the healing process [42,43,44,45,116,117]. The application of antimicrobial coatings has been explored to mitigate the occurrence of surgical-site infections, thereby facilitating wound closure and alleviating patient discomfort [42,43,44,45,116,117]. Consequently, our investigation focused on examining the inhibitory effects of biodegradable, braided, dip-coated PGA surgical sutures (S) (Table 5). Notably, the highest susceptibility to the tested biohybrid on surgical sutures was observed in C. albicans WDCM 00054 with a ZOI of 15 mm, hereby confirming our formulation as a strong antifungal agent (Table 5, Figure 10). 

In comparison, AV-PVP-Thyme-I_2_ showed a very low ZOI of 3 mm against the same fungus [42]. 

In general, the inhibitory zones for the title formulation AV-PVP-Thymol-I_2_ on sutures are much higher than compared to the Thyme formulation. AV-PVP-Thymol-I_2_ exhibited a ZOI of 9 mm for the Gram-positive bacteria *S. aureus* ATCC 25923, followed by ZOI = 6 for *B. subtilis* WDCM 00003 and *E. faecalis* ATCC 29212 (Table 5, Figure 10b). Further inhibited Gram-positive bacteria include *S. pyogenes* ATCC 19615 and *S. pneumoniae* ATCC 49619, both with ZOI = 2 (Table 5). *K. pneumoniae* WDCM 00097 and *E. coli* WDCM 00013 are the only susceptible Gram-negative pathogens on dip-coated surgical sutures with a ZOI of 3 mm, while the rest are all resistant (Table 5, Figure 10).

In summary, our research suggests that absorbable, surgical PGA sutures treated with our unique formulation AV-PVP-Thymol-I_2_ exhibit promise as antimicrobial agents in the prevention of surgical-site infections caused by *C. albicans* WDCM 00054, Gram-positive bacteria E. faecalis ATCC 29212, S. aureus ATCC 25923, and B. subtilis WDCM 00003, as well as Gram-negative bacteria *K. pneumoniae* WDCM 00097 and *E. coli* WDCM 00013.

Face masks play a crucial role as personal protective equipment in curbing the transmission of microbes, not only during typical flu or cold seasons but also in various public settings including patient care facilities [8,42,43,118,119,120,121,122,123,124]. During the COVID-19 pandemic, face masks were identified as indispensable tools in combating opportunistic pathogens [8,42,43,118,119,120,121,122,123,124]. The recent COVID-19 outbreak underscored the global challenge of providing high-quality face masks to the public [8,42,43,118,119,120,121,122,123,124]. Supply chains faltered, pharmacies and drug stores experienced shortages, and prices surged. The quality and safety of face masks were questioned, and they added to environmental pollution in the form of huge waste piles all over the world [8,42,43,118,119,120,121,122,123,124]. Employing methods such as reusing face masks by treating them with antimicrobial surface agents can help mitigate microbial proliferation. Subsequently, such measures would reduce environmental pollution, enable the safety of low-income populations, and promote sustainability.

We applied the title formulation AV-PVP-Thymol-I_2_ to both surgical face masks (M) and KN95 masks, investigating its inhibitory effects (Table 5). Our findings verify the previously observed patterns across different pathogens. The highest inhibitory action was demonstrated against *C. albicans* WDCM 00054, once again verifying the very strong antifungal action with ZOI = 66, 84, and 80 mm for blue (M^B^) and white (M^W^) mask layers and KN95 masks, respectively (Table 5, Figure 11). 

These are followed by a notable inhibition of Gram-positive bacteria *S. aureus* ATCC 25923 (ZOI = 40, 36, 32) and *B. subtilis* WDCM 00003 (ZOI = 30, 36, 30) for M^B^, M^W^, and KN95, respectively (Table 5). High antimicrobial activities are also noted against the other Gram-positive bacteria (Table 5). Meanwhile, elevated susceptibility is observed against Gram-negative bacteria *K. pneumoniae* WDCM 00097 (26, 23, 30 mm), *E. coli* WDCM 00013 (26, 35, 23 mm), and *P. aeruginosa* WDCM 00026 (13, 16, 18 mm) for M^B^, M^W^, and KN95, respectively (Table 5). 

Similar but more modest results were observed for the entire range of pathogens for the Thyme formulation AV-PVP-Thyme-I_2_ [42]. Moreover, the inhibitory effects against C. albicans WDCM 00054 are notably improved when AV-PVP-Thyme-I_2_ is incorporated into KN95 masks (80 mm), followed by the white surgical face mask layer (55 mm) and the blue layer (45 mm) [42]. 

In conclusion, AV-PVP-Thymol-I_2_ demonstrates potent inhibitory effects against various microorganisms on both surgical face masks and KN95 face masks. These include the fungal strain *C. albicans* WDCM 00054 and Gram-positive bacteria such as *S. aureus* ATCC 25923 and *B. subtilis* WDCM 00003 (Table 5). Additionally, Gram-negative pathogens like *K. pneumoniae* WDCM 00097, E. coli WDCM 00013, and P. aeruginosa WDCM 00026 exhibit heightened susceptibility when tested on surgical face masks and KN95 masks in comparison to the impregnated discs. These results are in accordance with the previously studied formulation AV-PVP-Thyme-I_2_ [42]. Nevertheless, the latter exhibits lessened inhibitory action against some pathogens. However, *P. mirabilis* ATCC 29906 is exclusively inhibited when AV-PVP-Thyme-I_2_ is impregnated onto KN95 masks, while it is resistant completely to AV-PVP-Thymol-I_2_ [42].

Sophisticated approaches to wound treatment are required to accelerate the healing process effectively while mitigating infection sources in the surrounding areas [42,43,95,98,99,100,101,102,103,125,126]. Cotton bandages soaked with antimicrobial substances can potentially curb the proliferation of opportunistic pathogens on the wound and shorten the duration of treatment [98,101,125]. Thus, we impregnated cotton bandages (B) with the title formulation AV-PVP-Thymol-I_2_ to investigate its efficacy against the studied strains (Table 5). As anticipated, AV-PVP-Thymol-I_2_ followed a similar pattern of antimicrobial activity. *B. subtilis* WDCM 00003 (ZOI = 40 mm) exerts the highest susceptibility towards AV-PVP-Thymol-I_2_, followed by *C. albicans* WDCM 00054 and *S. aureus* ATCC 25923, both with ZOI = 30 mm (Table 5, Figure 12). 

Gram-positive and Gram-negative pathogens exhibit inhibition zones around 23, 22, and 21 mm (Table 5). In comparison, similar patterns were observed in our previous study on AV-PVP-Thyme-I_2_, except *C. albicans* WDCM 00054 demonstrated the largest inhibition zone (53 mm) [42]. 

As a result, AV-PVP-Thymol-I_2_-impregnated cotton bandages, sutures, face masks, and KN95 masks have the potential to be used in mitigating microbial infections. In general, the inhibitory action is highest against *C. albicans* WDCM 00054, followed by Gram-positive and Gram-negative pathogens.

The overall susceptibility pattern remains consistent across all impregnated materials such as discs, sutures, cotton bandages, surgical face masks, and KN95 masks. Accordingly, the highest to lowest susceptibility, starting with *C. albicans* WDCM 00054, followed by Gram-positive and, finally, Gram-negative microorganisms, is presented in Table 5 in descending order. Bacterial morphology, size, and motility determine the susceptibility towards AV-PVP-Thymol-I_2_. Nevertheless, depending on the impregnated material (disc, suture, cotton bandage, KN95 mask, and surgical face mask), significant inhibition zones are observed. Bio-compounds from AV and Thymol penetrate the cell membranes and initiate cell death. However, the key factor remains the release of triiodide ions from the sustained-release reservoir PVP. These moieties decompose to free molecular iodine, which is a small, lipid-soluble molecule, able to diffuse through the lipophilic peptidoglycan layers of Gram-positive pathogens, facilitating cell death. Gram-negative pathogens are less affected by molecular iodine, Thymol, and AV biocomponents. Porin channels in Gram-negative bacteria allow the diffusion of hydrophilic triiodide ions, iodide ions, and small phenolic acids through the outer membranes. 

The susceptibility towards AV-PVP-Thymol-I_2_ is determined by the microbial morphology in accordance with our previous investigations [42,43,45,46]. The Gram-positive pathogen *S. aureus* ATCC 25932, characterized by clusters of immotile cocci, is highly inhibited, followed by immotile chains (*S. pyogenes* ATCC 19615, *E. faecalis* ATCC 29212) and pairs of cocci (*E. faecalis* ATCC 29212, *S. pneumoniae* ATCC 49619) (Figure 13). 

In contrast, the inhibition of Gram-negative bacteria correlates with their motility. The most active, swarming bacteria, *P. mirabilis* ATCC 29906, is only inhibited on KN95 by AV-PVP-Thyme-I_2_ [42]. Nevertheless, even the motile Gram-negative bacteria were inhibited very well by the uniformly coated materials featuring small, spherical, cube-like patches of AV-PVP-Thymol-I_2_ on bandages, face masks, KN95, and sutures. The title bioformulation AV-PVP-Thymol-I_2_ has the same pattern of antimicrobial activity as our previous biohybrid AV-PVP-Thyme-I_2_ (Table 5) [42]. Both formulations contain Thymol as a major ingredient and therefore depict similar results. AV-PVP-Thymol-I_2_ is a strong antifungal on all tested materials except cotton bandages in comparison to AV-PVP-Thyme-I_2_ [42]. AV-PVP-Thymol-I_2_ showcases an inhibitory zone (ZOI) of 30 mm, while AV-PVP-Thyme-I_2_ has a ZOI of 53 mm [42]. The disc diffusion studies on sterile discs and PGA sutures (S) have generally higher ZOIs in the title formulation, AV-PVP-Thymol-I_2_, compared to AV-PVP-Thyme-I_2_ [42]. The results on surgical masks (M^W^/M^B^) show stronger inhibition of the reference strains by AV-PVP-Thymol-I_2_ [42]. Impregnated KN95 and bandages (B) achieve better inhibition in the Thyme formulation [42]. 

Storing the stock solution AV-PVP-Thymol-I_2_ for 25 months results in slightly reduced antimicrobial activities. In contrast, the Thyme-extract-based AV-PVP-Thyme-I_2_ shows enhanced antimicrobial properties against the same 10 pathogens after 18 months [42]. Consequently, Thyme extract biocomponents in AV-PVP-Thyme-I_2_, like Thymol, carvacrol and rosmarinic acid, are responsible (together with AV components) for sustaining the inhibitory action during its storage [42]. The title formulation AV-PVP-Thymol-I_2_ consists of Thymol alone in combination with AV biocomponents. During its storage for up to 25 months, the polymeric iodophor PVP-I_2_ slowly releases iodine due to the lack of key ingredients carvacrol, rosmarinic acid, and others originating from the natural Thyme extract [42]. Thymol alone cannot sustain the sustained-release reservoir. Additionally, Thymol has strong hydrogen bonding properties and, over time, replaces triiodide ions on the PVP backbone, thus slowly reducing the inhibitory action of AV-PVP-Thymol-I_2_ (Figure 14) [79]. 

As a conclusion, the obtained data from the antimicrobial testing of AV-PVP-Thymol-I_2_ on discs, surgical sutures, bandages, face masks, and KN95 masks are promising, showing its successful use as an antimicrobial agent. AV-PVP-Thymol-I_2_ has the potential to mitigate and reduce inflammation as well as wound and surgical-site infections. It can be promising as a surface-killing agent and in reducing the airborne transmission of microbes. Further investigations are planned to verify its potential uses through in vivo and toxicity studies.

## 3. Materials and Methods

### 3.1. Materials

In December, we collected AV leaves from the botanical garden situated at the Ajman University campus in Ajman, UAE. Reference strains including *E. coli* WDCM 00013 Vitroids, *K. pneumoniae* WDCM 00097 Vitroids, *P. aeruginosa* WDCM 00026 Vitroids, *B. subtilis* WDCM 0003 Vitroids, and *C. albicans* WDCM 00054 Vitroids were procured from Sigma-Aldrich Chemical Co. in St. Louis, MO, USA. Additionally, *P. mirabilis* ATCC 29906, *S. aureus* ATCC 25923, *S. pyogenes* ATCC 19615, *E. faecalis* ATCC 29212, and *S. pneumoniae* ATCC 49619 were obtained from Liofilchem (Roseto degli Abruzzi, TE, Italy). Sigma Aldrich also supplied Mueller Hinton Broth (MHB), Sabouraud Dextrose broth, Thymol, ethanol, iodine (≥99.0%), and polyvinylpyrrolidone (PVP-K-30). Disposable sterilized Petri dishes containing Mueller Hinton II agar, McFarland standard sets, and antibiotic discs of gentamicin (9125, 30 µg/disc) and nystatin (9078, 100 IU/disc) were provided by Liofilchem Diagnostici based in Roseto degli Abruzzi (TE), Italy. Sterile filter paper discs with 6 mm diameters were procured from Himedia located in Jaitala Nagpur, Maharashtra, India. Sterile cotton gauze bandages, surgical disposable 3-ply, non-woven face masks, and KN95 face masks were obtained from a local pharmacy (FOMED, Qianjiang City, China). Sterile polyglycolic acid (PGA) surgical sutures (USP: 3-0, Metric: 2, 19 mm, 75 cm, DC3K19) were provided by General Medical Disposable (GMD), GMD Group A.S., Istanbul, Turkey. All reagents used were of analytical grade and employed under sterile conditions, with absolute ethanol and ultrapure water being utilized in all experiments. 

### 3.2. Preparation of Aloe vera (AV) Extract

The *Aloe vera* (*A. barbadensis* Miller) leaves were harvested in December from the botanical garden of Ajman University during the early morning hours [42,43,45,46]. Within a span of 10 min, leaves measuring 35 to 50 cm were carefully washed with water and then rinsed thoroughly with distilled water, then absolute ethanol, and subsequently rinsed multiple times with ultrapure water. They were then allowed to air dry for 1 h. Once dried, the AV leaves were cut with a sterile knife to facilitate the collection of the mucilaginous gel into a sterile beaker. The gel was then transferred to a sterile mixer and homogenized for 10 min until a consistent texture was achieved. Afterwards, the AV gel was centrifuged for 40 min at 4000 rpm (3K 30; Sigma Laborzentrifugen GmbH, Osterode am Harz, Germany). The resulting supernatant, possessing a light-yellow hue, was promptly transferred into a sterile brown screw-capped bottle and placed in a refrigerator at 3 °C until further use.

### 3.3. Preparation of AV-PVP-Thymol-I_2_

AV-PVP-Thymol-I_2_ was synthesized through a straightforward one-pot process. Firstly, 2 mL of pure AV gel was dispensed into a sterile beaker. Subsequently, 2 mL of a recently prepared solution containing 1 g of polyvinylpyrrolidone K-30 (PVP) dissolved in 10 mL of distilled water was introduced into the beaker under stirring at ambient temperature. Following this, 2 mL of Thymol solution (0.15 g in 10 mL of ethanol, 100 µg/mL) was slowly added to the mixture while stirring. Finally, 2 mL of a freshly prepared iodine solution (0.05 g of iodine in 3 mL of absolute ethanol) was incorporated into the mixture under continuous stirring and at room temperature. The resulting formulation, AV-PVP-Thymol-I_2_, was promptly transferred into a screw-capped sterile glass sample tube and stored in darkness at 3 °C in a refrigerator for subsequent use.

### 3.4. Characterization of AV-PVP-Thymol-I_2_

The purity and formulation morphology of AV-PVP-Thymol-I_2_ were confirmed through analysis utilizing SEM/EDS, Raman Spectroscopy, UV-vis, FTIR, and X-ray diffraction (XRD).

#### 3.4.1. Scanning Electron Microscopy (SEM) and Energy-Dispersive X-ray Spectroscopy (EDX)

The VEGA3 from TESCAN (Brno, Czech Republic) was utilized for scanning electron microscopy (SEM) and energy-dispersive X-ray spectroscopy (EDS) examination, operating at 15 kV. A solution of the AV-PVP-Thymol-I_2_ formulation was diluted with distilled water, then applied onto a carbon-coated copper grid and subsequently dried. Following this, a gold coating was applied using the Quorum Technology Mini Sputter Coater (Brno, Czech Republic). SEM analysis provided insights into the morphology, while EDS analysis verified the purity of the formulation AV-PVP-Thymol-I_2_.

#### 3.4.2. UV-Vis Spectrophotometry (UV-Vis)

The analysis of AV-PVP-Thymol-I_2_ was conducted using a UV-Vis spectrophotometer, specifically the model 2600i manufactured by Shimadzu in Kyoto, Japan. Measurements were taken across the wavelength spectrum from 195 to 800 nm.

#### 3.4.3. Raman Spectroscopy 

The composition AV-PVP-Thymol-I_2_ underwent analysis under ambient conditions using a RENISHAW system located in Gloucestershire, UK, featuring an optical microscope. The sample was introduced into a cuvette measuring 1 cm × 1 cm and positioned in front of the laser beam, which had an excitation wavelength of 785 nm. Utilizing a 50× magnification confocal microscope, the beam was focused to a spot diameter of 2 microns onto the sample. Scattered light was captured by a CCD-based monochromator with a spectral range spanning 50–3400 cm^−1^ and a spectral resolution of −1 cm^−1^. The monochromator had an output power of 0.5% and an integration time of approximately 30 s.

#### 3.4.4. Fourier-Transform Infrared Spectroscopy (FTIR)

The AV-PVP-Thymol-I_2_ formulation was subjected to Fourier-Transform Infrared (FTIR) analysis within the spectral range of 400 to 4000 cm^−1^ using an Attenuated Total Reflectance (ATR) IR spectrometer equipped with a Diamond window (Shimadzu, Kyoto, Japan).

#### 3.4.5. X-ray Diffraction (XRD)

AV-PVP-Thymol-I_2_ underwent X-ray diffraction analysis using a BRUKER instrument (D8 Advance, located in Karlsruhe, Germany) equipped with Cu radiation at a wavelength of 1.54060 Å. The analysis employed a Two Theta configuration, with a time per step of 0.5 s and a step size of 0.03.

### 3.5. Bacterial Strains and Culturing

Antimicrobial testing was conducted on the title compound AV-PVP-Thymol-I_2_ against a panel of 10 standard microbial strains, including *C. albicans* WDCM 00054 Vitroids, *S. aureus* ATCC 25923, *S. pneumoniae* ATCC 49619, *E. faecalis* ATCC 29212, *S. pyogenes* ATCC 19615, *B. subtilis* WDCM 0003 Vitroids, *K. pneumoniae* WDCM 00097 Vitroids, *E. coli* WDCM 00013 Vitroids, *P. aeruginosa* WDCM 00026 Vitroids, and *P. mirabilis* ATCC 29906. These strains were initially stored at −20 °C and later revived by inoculating fresh microbes into Mueller Hinton Broth (MHB) with a concentration of 1 × 10^6^ CFU/mL (OD_600_ = 0.02), followed by refrigeration at 4 °C until use [42,43,44,45,46].

### 3.6. Determination of Antimicrobial Activities of AV-PVP-Thymol-I_2_

AV-PVP-Thymol-I_2_ underwent testing against a panel of 10 reference strains, including *C. albicans* WDCM 00054 Vitroids, *S. aureus* ATCC 25923, *S. pneumoniae* ATCC 49619, *E. faecalis* ATCC 29212, *S. pyogenes* ATCC 19615, *B. subtilis* WDCM 0003 Vitroids, *K. pneumoniae* WDCM 00097 Vitroids, *E. coli* WDCM 00013 Vitroids, *P. aeruginosa* WDCM 00026 Vitroids, and *P. mirabilis* ATCC 29906. Gentamicin and nystatin served as positive controls, the latter specifically for C. albicans WDCM 00054 Vitroids. Negative controls included pure ethanol and ultrapure water, neither of which exhibited inhibitory effects. Each test was conducted thrice, and the average results were reported. Additionally, the formulation was applied to sterile discs, cotton gauze bandages, sterile PGA sutures, KN95 masks, and surgical facemasks, to be tested against the same reference strains. 

#### 3.6.1. Procedure for Zone of Inhibition Plate Studies

We employed the zone of inhibition plate method to assess the susceptibility of 10 microbial strains to AV-PVP-Thymol-I_2_ in accordance with our previous investigations [42,43,44,45,46,127]. The fungus *C. albicans* WDCM 00054 was cultured at 30 °C in Sabouraud Dextrose broth. The chosen bacterial reference strains were suspended in 10 mL of Mueller Hinton broth (MHB) and then incubated for 2 to 4 h at 37 °C [42,43,44,45,46]. All microbial cultures were readily adjusted to a 0.5 McFarland standard. Disposable, sterilized Petri dishes were evenly seeded with 100 μL of microbial culture using sterile cotton swabs [42,43,44,45,46]. After allowing the plates to dry for 10 min, they were prepared for antimicrobial testing of AV-PVP-Thymol-I_2_.

#### 3.6.2. Disc Diffusion Method (DD)

We adhered to the guidelines set forth by the Clinical and Laboratory Standards Institute (CLSI) for conducting antimicrobial testing [128]. Filter paper discs were sterilized and then soaked in AV-PVP-Thymol-I_2_ solutions of various concentrations (11 µg/mL, 5.5 µg/mL, 2.75 µg/mL, and 1.38 µg/mL) for 24 h [42,43,44,45,46]. Subsequently, these discs were air-dried for 24 hours under normal environmental conditions [42,43,44,45,46]. Gentamycin and nystatin antibiotic discs served as positive controls. The diameter of the inhibition zone (ZOI), indicating the clear area surrounding the disc, was measured using a ruler to the nearest millimetre. The absence of a discernible inhibition zone indicated a lack of inhibition against the reference microbial strain.

### 3.7. Preparation and Analysis of Impregnated Sutures, Cotton Gauze Bandages, KN95 Masks, and Surgical Face Masks with AV-PVP-Thymol-I_2_ (11 µg/mL)

The aseptic, woven surgical PGA sutures, measuring 2.5 mL in length, underwent a 24 h immersion in a 50 mL solution of AV-PVP-Thymol-I_2_ at room temperature. After immersion, the originally blue sutures transitioned to a brownish-blue hue and were subsequently air-dried for 24 h under standard conditions [42,43,44,45,46]. Cotton gauze bandages, KN95 masks, and surgical face masks were aseptically cut into square pieces measuring 5 cm × 5 cm using sterile scissors [42,43,44,45,46]. These squares were then immersed in 50 mL of AV-PVP-Thymol-I_2_ solution for 24 h and air-dried for an additional 24 h at room temperature [42,43,44,45,46]. The formulation had a concentration of 11 µg/mL [42,43,44,45,46].

Subsequently, all impregnated materials underwent testing using disc diffusion methods against the same consistent set of 10 reference microbial strains in accordance with our previous investigations [42,43,44,45,46,127,128].

### 3.8. Statistical Analysis

SPSS software (version 17.0, SPSS Inc., Chicago, IL, USA) was employed for statistical analysis, with data presented in mean values. The significance between groups was determined through one-way ANOVA. Statistical significance was defined as *p* < 0.05.

## 4. Conclusions

Antimicrobial resistance poses an ongoing, dangerous challenge to humankind. Microorganisms adapt various mechanisms to escape efforts aimed at controlling their spread. Multi-drug-resistant ESKAPE pathogens are already escalating rates of illness and death worldwide. Novel, cost-effective, sustainable, and easily implementable strategies are pivotal to address this issue.

A promising avenue includes investigating natural antimicrobial formulations with synergetic bio-compounds within natural plant extracts. Plants develop comprehensive defence mechanisms through the synergistic action of bio-compounds. We combined AV gel and Thymol within a sustained-release reservoir of PVP-I_2_. The microbicide iodine, complexed into PVP, enhances the antimicrobial properties of the plant-derived materials from AV and Thymol. The resulting formulation, AV-PVP-Thymol-I_2_, has demonstrated potent antifungal activity against *C. albicans* WDCM 00054 and the selection of Gram-positive, as well as Gram-negative, microorganisms when uniformly applied to sterile discs, bandages, sutures, and surgical and KN95 face masks. The SEM of the coated material shows a homogenous surface with small patches, which results in high inhibitory action against the selected pathogens.

The title formulation still exerts inhibitory action after a storage period of 25 months, but at lower rates. However, in comparison to AV-PVP-Thymol-I_2_, the title compound AV-PVP-Thymol-I_2_ exhibited two unexpected properties. Firstly, the incorporation of Thymol instead of Thyme extract enhanced the antimicrobial properties of the title formulation AV-PVP-Thymol-I_2_ strongly against the selection of microorganisms. Secondly, storage for more than two years (25 months) reduces the susceptibility of the reference strains against AV-PVP-Thymol-I_2_ slowly, while the opposite happened to AV-PVP-Thyme-I_2_. Using Thymol instead of a Thyme extract has the advantage of much higher inhibitory action, but this property slowly decreases over a storage time of 2 years. Consequently, Thyme-extract-mediated antimicrobial activity is a result of synergetic bio-compounds and therefore stabilizes the sustained-release reservoir. Meanwhile, Thymol in AV-PVP-Thymol-I_2_ slowly competes over a span of 2 years with the triiodide ions, resulting in their premature release.

The incorporation of AV-PVP-Thymol-I_2_ into face masks exhibited significant inhibition against nine reference strains except for *P. mirabilis* ATCC 29906. Nevertheless, the results are enough to suggest the potential for sustainable mask reuse, benefiting both the environment and economically disadvantaged communities. Additionally, AV-PVP-Thymol-I_2_ could be utilized as a skin antiseptic or disinfectant for surfaces. The latter could mitigate both community and hospital-acquired, nosocomial infections. Moreover, the promising results observed on bandages support the integration of AV-PVP-Thymol-I_2_ into wound-dressing materials. Further research is necessary to assess its biological activities in vivo, as well as its cytotoxicity and potential side effects.

## Figures and Tables

**Figure 1 ijms-25-04949-f001:**
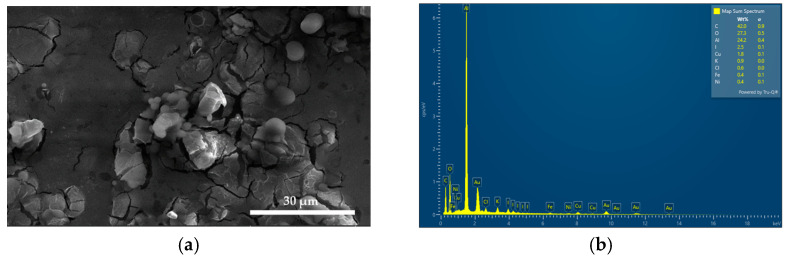
AV-PVP-Thymol-I_2_ analysis: (**a**) scanning electron microscopy (SEM); (**b**) energy-dispersive spectroscopy (EDS).

**Figure 2 ijms-25-04949-f002:**
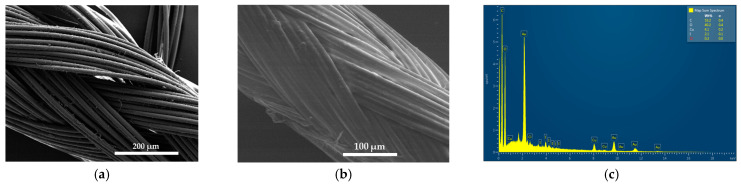
SEM of surgical sutures coated with AV-PVP-Thymol-I_2_: (**a**) plain PGA suture, 200 µm [45]; (**b**) coated PGA suture, 100 µm; (**c**) EDS.

**Figure 3 ijms-25-04949-f003:**
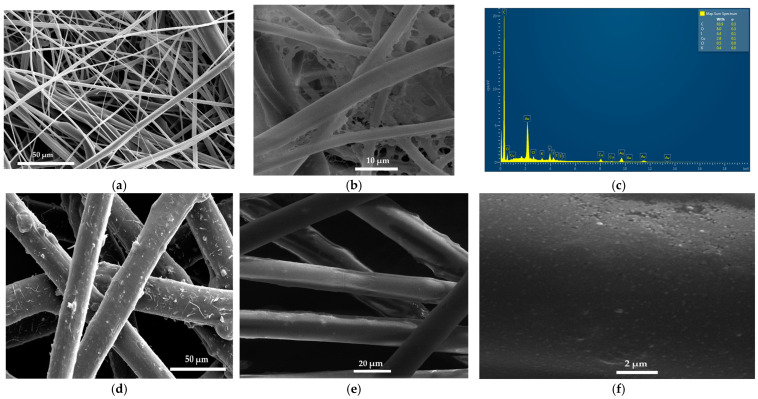
SEM of surgical face masks impregnated with AV-PVP-Thymol-I_2_. Dense, white inner layer: (**a**) plain at 50 µm; (**b**) coated at 10 µm; (**c**) EDS. Network-like, blue outer layer: (**d**) plain at 50 µm; (**e**) coated at 20 µm; (**f**) detailed view at 2 µm.

**Figure 4 ijms-25-04949-f004:**
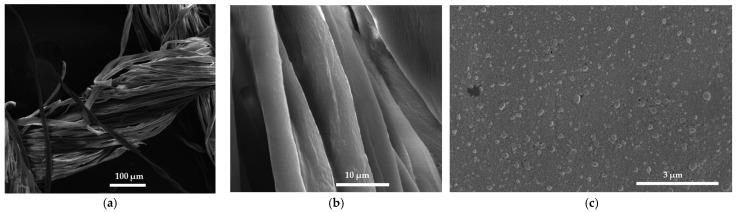
SEM of cotton surgical bandages impregnated with AV-PVP-Thymol-I_2_: (**a**) 100 µm; (**b**) 10 µm; (**c**) detailed view at 3 µm.

**Figure 5 ijms-25-04949-f005:**
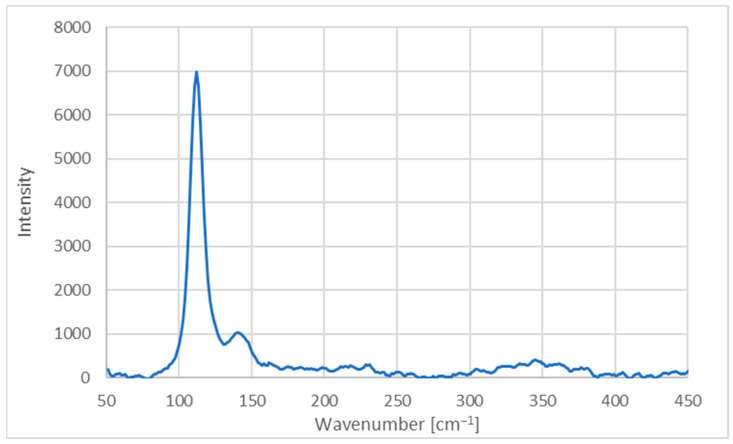
Raman spectroscopic analysis of AV-PVP-Thymol-I_2_.

**Figure 6 ijms-25-04949-f006:**
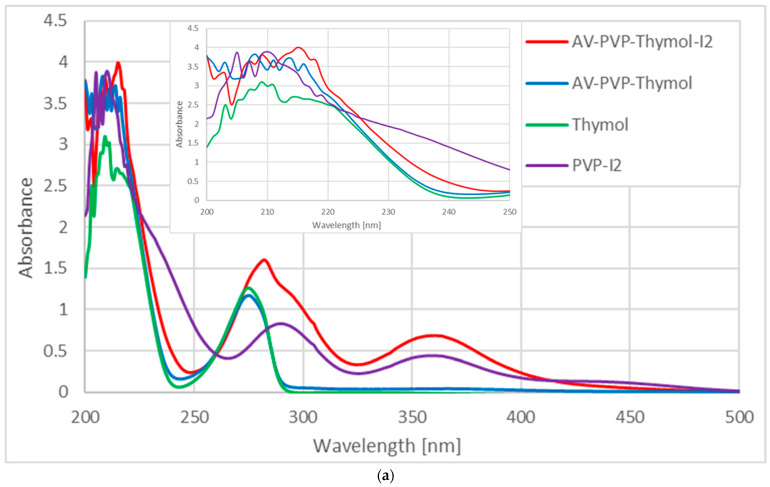
UV-vis analysis of AV-PVP-Thymol-I_2_, AV-PVP-Thymol-I_2_ (after 25 months), AV-PVP-Thymol, PVP-I_2_, and Thymol (200–500 nm). (AV-PVP-Thymol-I_2_: red; AV-PVP-Thymol-I_2_ after 25 months of storage: light blue; AV-PVP-Thymol: dark blue; PVP-I_2_: purple; Thymol: green.): (**a**) fresh samples; (**b**) comparison between AV-PVP-Thymol-I_2_, AV-PVP-Thymol-I_2_ (after 25 months).

**Figure 7 ijms-25-04949-f007:**
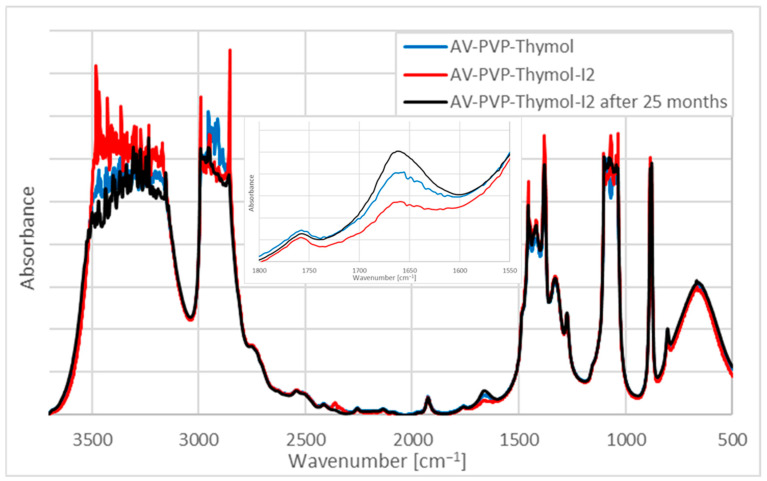
Fourier-Transform Infrared (FTIR) spectroscopic analysis of AV-PVP-Thymol-I_2_ (red), 25-month-old AV-PVP-Thymol-I_2_ (black), and AV-PVP-Thymol (blue).

**Figure 8 ijms-25-04949-f008:**
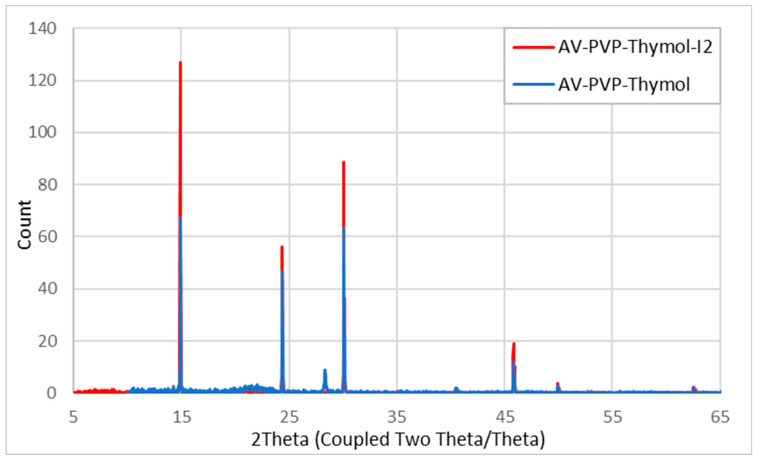
X-ray diffraction (XRD) analysis of AV-PVP-Thymol-I_2_ (red) and AV-PVP-Thymol (blue).

**Figure 9 ijms-25-04949-f009:**
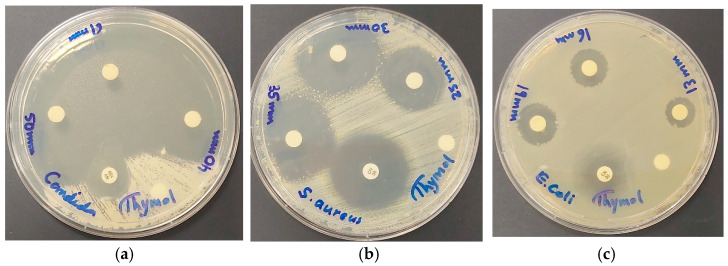
AV-PVP-Thymol-I_2_-coated sterile discs (disc diffusion assay with concentrations of 11, 5.5, and 2.75 µg/mL) with positive control antibiotics nystatin (100 IU) and gentamicin (30 µg/disc). From left to right: AV-PVP-Thymol-I_2_ against (**a**) *C. albicans* WDCM 00054; (**b**) *S. aureus* ATCC 25932; (**c**) *E. coli* WDCM 00013.

**Figure 10 ijms-25-04949-f010:**
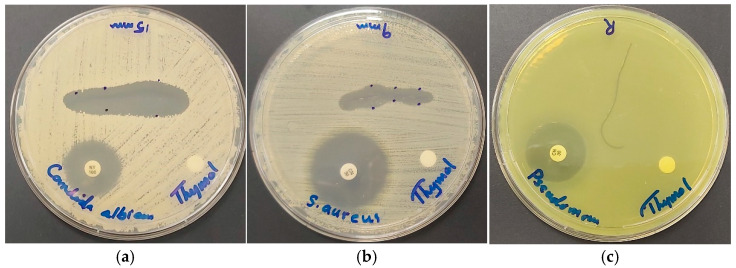
AV-PVP-Thymol-I_2_-coated sterile PGA sutures with positive control antibiotics nystatin (100 IU) and gentamicin (30 µg/disc). From left to right: AV-PVP-Thymol-I_2_ (11 µg/mL) against (**a**) *C. albicans* WDCM 00054; (**b**) *S. aureus* ATCC 25932; (**c**) *P. aeruginosa* WDCM 00026.

**Figure 11 ijms-25-04949-f011:**
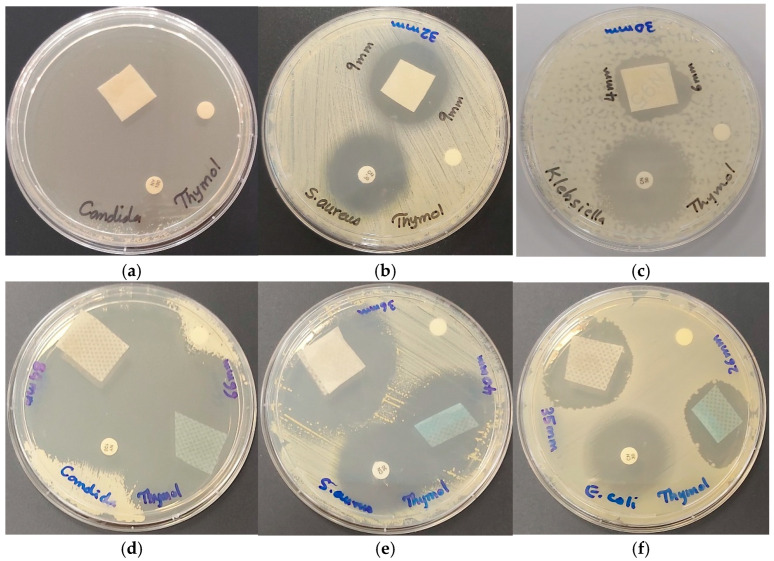
AV-PVP-Thymol-I_2_-coated sterile mask tissues with positive control antibiotics nystatin (100 IU) and gentamicin (30 µg/disc). From left to right: AV-PVP-Thymol-I_2_ (11 µg/mL) on KN95 mask tissue: (**a**) *C. albicans* WDCM 00054; (**b**) *S. aureus* ATCC 25932; (**c**) *K. pneumoniae* WDCM 00097. AV-PVP-Thymol-I_2_ (11 µg/mL) on surgical face mask tissues (blue and white): (**d**) *C. albicans* WDCM 00054; (**e**) *S. aureus* ATCC 25932; (**f**) *E. coli* WDCM 00013.

**Figure 12 ijms-25-04949-f012:**
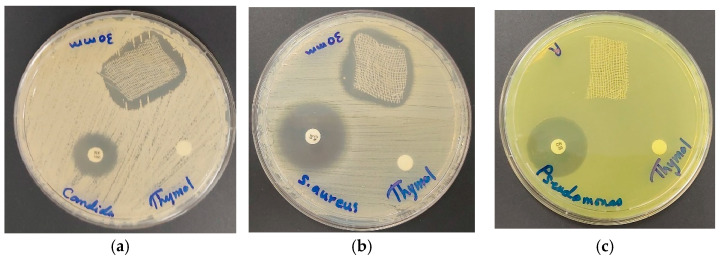
AV-PVP-Thymol-I_2_-coated sterile bandages with positive control antibiotics nystatin (100 IU) and gentamicin (30 µg/disc). From left to right: AV-PVP-Thymol-I_2_ (11 µg/mL) against (**a**) *C. albicans* WDCM 00054; (**b**) *S. aureus* ATCC 25932; (**c**) *P. aeruginosa* WDCM 00026.

**Figure 13 ijms-25-04949-f013:**
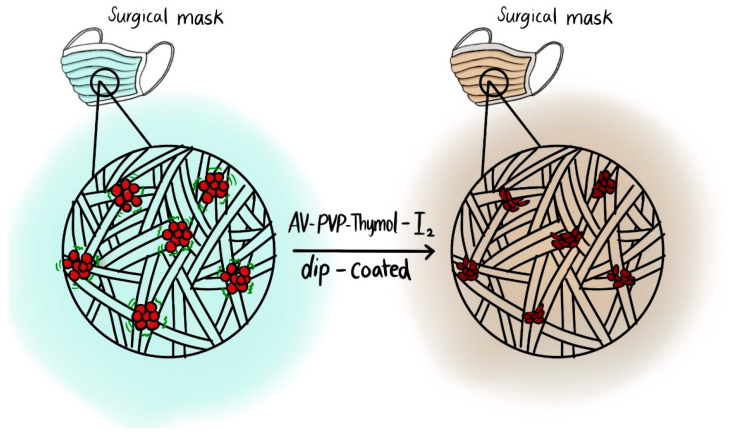
AV-PVP-Thymol-I_2_ coated on surgical face masks against *S. aureus* ATCC 25932.

**Figure 14 ijms-25-04949-f014:**
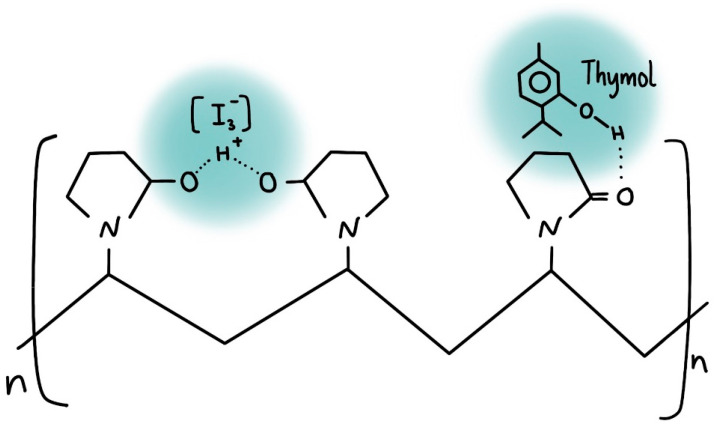
AV-PVP-Thymol-I_2_ and hydrogen bonding on PVP: competition between Thymol and triiodide ions.

**Table 1 ijms-25-04949-t001:** Raman shifts in AV-PVP-Thymol (1), AV-PVP-Thyme-I_2_ (2) [42], AV-PVP-Sage-I_2_ [43], and other related compounds (cm^−1^).

Group	AV-PVP-Thymol-I_2_	[42]	[43]	[29]	[38]	[20]
I_2_ [I_2_^….^I^−^]			sh, w 80 * s 169 * ν_as_ w 189 * ν	m 85 * m 160 * ν_as_		s 169ν_s_
I_3_^−^ **[I-I-I^−^]**	**s 112ν_1,s_**	**s 112ν_1,s_**	sh, w 61δ_def_ sh, w 70ν_2bend_ **vs 110ν_1,s_** vw 222^+^ 2ν_1,s_	sh 60δ_def_ sh, w 75ν_2bend_ **s 110ν_1,s_** vw 221 2ν_1,s_	**114ν_1,s_**	**vs 111ν_s_**
I_3_^−^ [I-I^….^I^−^]	w 141ν_3,as_ w 148ν_3,as_	w 141ν_3,as_ w 145ν_3,as_	m 144ν_3,as_	m 144ν_3,as_	144ν_3,as_	m 145ν_s_
	vw 345ν_as_		vw 334ν_as_	sh, vw 154ν_3,as_		

ν = vibrational stretching, _s_ = symmetric, _as_ = asymmetric, _1_ = stretching mode 1, _3_ = stretching mode 3, bend = bending, δ_def_ = deformation. * belong to the same asymmetric, nonlinear unit I_3_^−^ = I_2_^……^I^−^. ^+^ overtones of triiodide ions. w = weak, vw = very weak, br = broad, s = strong, vs = very strong, m = intermediate, sh = shoulder.

**Table 2 ijms-25-04949-t002:** UV-vis absorption signals in the samples AV-PVP-Thymol (1), AV-PVP-Thymol-I_2_ (2), AV-PVP-Thymol-I_2_ after 25 months (3), AV-PVP-Thyme-I_2_ [42], PVP-I_2_, Thymol, and AV-PVP-Sage-I_2_ [43] [nm].

Group	1	2	3	[42]	PVP-I_2_	Thymol	[43]
I_2_		203 vs	203 vs	204 vs	205 vs		206 vs
I_3_^−^ as [I-I-I^−^] I_3_^−^ as [I-I^….^I^−^]		292 s, sh, br 360 m, br	292 s, sh, br 360 m, br	289 s, br 359 s, br	290 m, br 360 m, br		290 s, br 359 s, br
I_5_^−^					444 w, br		
I^−^		201 s	201 sh, vs	201 sh, vs	202 sh		202 vs
AV/Aloin	208 vs	208 s	208 vs	208 vs 230 sh			206 vs
PVP	207 sh **213 s, sh**	207 s **213 s, sh**	208 vs	208 vs 209 sh **213 s,sh**	207 vs 210 vs 212 sh **215 sh**		201–205 ** 209 vs 211 br 215 sh
	216 s, sh 218 s, sh	216 s, sh **218 vs, sh**	216 s,sh **220 vs, sh**	217 sh	217 sh **219 s, sh**		
	221 vs, sh 224 m, sh	**222 vs, sh** 225 m, sh	**223 vs** 224 sh	224 sh	221 s, sh 224 s 231 sh		
PVP-I_2_	304 vw	304 m, sh	304 m, sh	304 w, sh	304 sh		305 s, sh
Thymol	203 vs 207 vs 211 vs 218 s, sh 276 m	**203 s** 204 vs **206 s** 207 vs **209 vs** 211 s 215 s **218 vs, sh** **250**–325 ** **282 s** **328**–440 **	**205 s** **209 s** **212 vs** **220 vs, sh** **250**–325 ** **282 s** **328**–440 **	202–220 ** 204 vs 206 s 207 vs 210 vs, sh **250**–320 ** 277 ** **330**–440 ** 415 br, vw		203 s 205 s 207 s 209 vs 210 vs 216 s 220 s 277 m	283 ** 340 m, sh

UV-vis absorption signals with concentration of 0.11 µg/mL. ** The broad bands overlap and several peaks related to AV compounds, iodine moieties, and thymol/carvacrol cannot be observed. vw = very weak, br = broad, s = strong, vs = very strong, m = intermediate, sh = shoulder.

**Table 3 ijms-25-04949-t003:** FTIR analysis of AV-PVP-Thymol-I_2_ (A), AV-PVP-Thymol (B), and 25-month-old AV-PVP-Thymol-I_2_ (C) in ethanol solvent [cm^−1^].

	Ν_1,2_ (O–H)_s,a_ ν (COOH)_a_	ν (C–H)_a_	ν (C-H)_a_	ν (C-H)_s_	ν (C=O)_a_	δ (C-H)_a_ δ (CH_2_) δ (O-H)	ν (C-C)	ν (C-O)	ν (C-O) ν (C-N)
A	3480 vs 3464 vs 3425 vs 3362 vs 3235 v 3169 vs 3152 s, sh	2990 vs	2949 vs (PVP)	2855 vs	1759 vw, br 1659 vw, br	1456 s δ(CH_3_)_s,in-plane_ 1420 s δ(CH_3_)_a,in-plane_ 885 s δ(CH_2_)_twisting_ 882 s δ(CH_2_)_twisting_ 878 s δ(CH_2_)_twisting_ 804 m δ(C-H)_out-of-plane_ 667 m δ(O-H)	1381 vs 1331 m, br	1275 m, br	1153 w, sh ν (C-N) 1101 vs ν (C-O) 1069 vs ν (C-O) 1051 s ν (C-O) 1042 s ν (C-O) 1037 br ν (C-O) 1036 s ν (C-O)
B	3480 vs 3464 vs 3425 vs 3362 vs 3235 vs 3169 vs 3152 s, sh	2990 vs	2953 s (PVP)	2855 s	1757 vw, br 1665 vw, br	1456 s δ(CH_3_)_s,in-plane_ 1420 s δ(CH_3_)_a,in-plane_ 882 s δ(CH_2_)_twisting_ 804 m δ(C-H)_out-of-plane_ 667 m δ(O-H)	1381 vs 1331 m, br	1275 m, br	1153 w, sh ν (C-N) 1095 s ν (C-O) 1071 vs ν (C-O) 1051 s ν (C-O) 1042 s ν (C-O) 1037 br ν (C-O)
C	3480 vs 3464 vs 3425 vs 3362 vs 3235 vs 3169 vs 3152 s, sh	2990 vs	2953 vs (PVP)	2855 vs	1757 vw, br 1661 vw, br	1456 s δ(CH_3_)_s,in-plane_ 1420 s δ(CH_3_)_a,in-plane_ 885 s δ(CH_2_)_twisting_ 882 s δ(CH_2_)_twisting_ 878 s δ(CH_2_)_twisting_ 804 m δ(C-H)_out-of-plane_ 667 m δ(O-H)	1381 vs 1331 m, br	1275 m, br	1153 w, sh ν (C-N) 1099 vs ν (C-O) 1071 vs ν (C-O) 1051 s ν (C-O) 1042 s ν (C-O) 1037 br ν (C-O) 1036 s ν (C-O)

ν = vibrational stretching, δ = deformation, _s_ = symmetric, _a_ = asymmetric; absorption intensity: vs = very strong, s = strong, m = medium, vw = very weak, sh = shoulder, br = broad, w = weak.

**Table 4 ijms-25-04949-t004:** XRD analysis of the samples AV-PVP-Thymol (1), AV-PVP-Thymol-I_2_ (2), AV-PVP-Thyme-I_2_ [42], AV-PVP-I_2_ [46], and in other investigations (2Theta^o^).

Group	AV-PVP- Thymol	AV-PVP- Thymol-I_2_	[42]	[46]	[83]	[40]	[35]
I_2_	-	-	-	-	-	25 29 36	24.5 s 25 s 28 s 37 w 38 w 43 w
							46 m
PVP	14.89 s 24.36 w	14.89 vs 24.37 w	14.89 s 24.37 w	10 s 19 s, br	-	-	-
Thymol	28.83 vw		14.92 s, br 14.98 m, br	-	11.8 w 15.8 w	-	-
	24.42 vw	24.43 vw	24.43 vw		16.6 vs		
	30.08 m	30.08 s	30.08 m		18.7 vs		
	30.14 w	30.14 w	30.14 vw		20.3 m 20.8 m		
					24 w 25.4 s		
AV	45.83 w 45.96 vw 49.95 vw 50.07 vw 62.53 vw 62.71 vw	45.84 w 45.96 vw 49.95 vw 50.07 vw 62.54 vw 62.72 vw	45.84 vw 45.95 vw 49.95 vw 62.57 vw	14 s 21 s, br 22 s, br	38.2 vs 44.4 m 64.9 w	-	-

w = weak, br = broad, s = strong, m = intermediate.

**Table 5 ijms-25-04949-t005:** Antimicrobial testing by disc dilution studies of antibiotics (A), AV-PVP-Thymol-I_2_ (1^+^,2^+^,3^+^), suture (S), bandage (B), mask blue layer (M^B^), mask white layer (M^W^), and KN95 mask. ZOI (mm) against microbial strains by diffusion assay.

Strain	Antibiotic	A	1^+^	2^+^	3^+^	S	B	M^B^	M^W^	KN95
*C. albicans* WDCM 00054	NY	16	61	50	40 *	15	30	66	84	80+
*S. aureus* ATCC 25923	G	28	35	30	25	9	30	40	36	32
*B. subtilis* WDCM 00003	G	21	32	24	19	6	40	30	36	30
*S. pyogenes* ATCC 19615	G	25	27	21	15	2.1	23	23	23	24
*E. faecalis* ATCC 29212	G	25	22	18	15	6	21	23	32	21
*S. pneumoniae* ATCC 49619	G	18	24	23	15	2.1	23	20	23	25
*K. pneumoniae* WDCM 00097	G	30	17	15	13	3	22	26	23	30
*E. coli* WDCM 00013	G	23	19	16	13	3	22	26	35	23
*P. aeruginosa* WDCM 00026	G	23	15	13	11	0	0	13	16	18
*P. mirabilis* ATCC 29906	G	30	0	0	0	0	0	0	0	0

^+^ Disc diffusion studies (6 mm disc impregnated with 2 mL of 11 µg/mL (1^+^), 2 mL of 5.5 µg/mL (2^+^), and 2 mL of 2.75 µg/mL (3^+^) of AV-PVP-Thymol-I_2_. A = Gentamicin (G, 30 µg/disc). Nystatin (NY, 100 IU). Suture (S), bandage (B), and mask (M) impregnated with 2 mL of 11 µg/mL AV-PVP-Thymol-I_2_. The grey shaded area represents Gram-negative bacteria. 0 = Resistant. * Further dilution to 1.38 µg/mL yielded ZOI = 19 mm. No statistically significant differences (*p* > 0.05) between row-based values through Pearson correlation.

**Table 6 ijms-25-04949-t006:** Antimicrobial testing by disc dilution studies of antibiotics (A), AV-PVP-Thymol-I_2_ in methanol (Me), AV-PVP-Thymol-I_2_ (1^+^), AV-PVP-Thymol-I_2_ (long-term stability study after 25 months) (L25), and AV-PVP-Thyme-I_2_ (long-term stability study after 18 months) [42]. Controls: PVP-I_2_ (11 µg/mL), AV-PVP-I_2_ (50 µg/mL) [46], and Thymol in ethanol (100 µg/mL) (T). ZOI (mm) against microbial strains by diffusion assay.

Strain	Antibiotic	A	Me	1^+^	L25	[42]	PVP-I_2_	[46]	T
*C. albicans* WDCM 00054	NY	16	72	61	30	60+	27	56	45
*S. aureus* ATCC 25923	G	28	19	35	14	22	13	25	43
*B. subtilis* WDCM 00003	G	21	10	32	11	19	13	22	40
*S. pyogenes* ATCC 19615	G	25	16	27	10	16	12	16	29
*E. faecalis* ATCC 29212	G	25	16	22	10	15	13	17	23
*S. pneumoniae* ATCC 49619	G	18	14	24	9	15	13	14	0
*K. pneumoniae* WDCM 00097	G	30	12	17	9	14	8	23	39
*E. coli* WDCM 00013	G	23	12	19	9	14	10	18	34
*P. aeruginosa* WDCM 00026	G	23	0	15	0	8	0	13	0
*P. mirabilis* ATCC 29906	G	30	0	0	0	0	0	0	0

^+^ Disc diffusion studies (6 mm disc impregnated with 2 mL of 11 µg/mL) of AV-PVP-Thymol-I_2_ in methanol as solvent (Me), AV-PVP-Thymol-I_2_ title formulation in ethanol as solvent (1^+^), AV-PVP-Thymol-I_2_ (long-term stability study of the title formulation in ethanol after 25 months) (L25), and AV-PVP-Thyme-I_2_ (long-term stability study after 18 months) [42]. Controls (6 mm disc impregnated with 2 mL): PVP-I_2_ (11 µg/mL), AV-PVP-I_2_ (50 µg/mL) [46], Thymol in ethanol (100 µg/mL) (T), and A = Gentamicin (G, 30 µg/disc). Nystatin (NY, 100 IU). The grey shaded area represents Gram-negative bacteria. 0 = Resistant. No statistically significant differences (*p* > 0.05) between row-based values through Pearson correlation.

## Data Availability

Data are contained within the article.

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
