# Peer review of "Thymol, a Monoterpenoid within Polymeric Iodophor Formulations and Their Antimicrobial Activities"

_ijms, 2024, doi:10.3390/ijms25094949_

Round 1
Reviewer 1 Report
Comments and Suggestions for Authors
Article Title: Thymol and polymeric iodophor formulations: Investigating antimicrobial activities
Authors: Zehra Edis, Samir Haj Bloukh
Journal: International Journal of Molecular Sciences
Brief Summary: the study explores the effectiveness of Thymol within a polymeric iodophor formulation (AV-PVP-Thymol-I2) as an antimicrobial agent against various pathogenic microorganisms. The paper outlines the preparation and characterization of the formulation and assesses its inhibitory effects on selected reference strains through multiple analytical techniques, including SEM/EDS, UV-vis, Raman, FTIR, and XRD analyses. The primary strengths of the paper are its detailed experimental design and comprehensive analysis, which support the potential of AV-PVP-Thymol-I2 as a promising antimicrobial agent.
General concept comments:
Areas of Weakness: the paper occasionally lacks specific details about the concentrations used in different formulations, particularly in Table 5's captions. This omission could lead to ambiguities in replicating the study or assessing the antimicrobial effectiveness accurately.
The distinction between the effects of Thymol and Thyme within the formulations is not clearly delineated, which could be misleading, as seen in Figure 11 where both are mentioned but only Thymol is shown.
Testability of the Hypothesis: the hypothesis that Thymol incorporated in AV-PVP-I2 can enhance antimicrobial properties is well-framed and tested through rigorous methodologies. However, the study would benefit from additional controls, such as testing the formulation against a broader range of bacteria and fungi to validate the spectrum of activity.
Methodological Inaccuracies: the paper should address the stability of the formulation over time, as the long-term efficacy of the antimicrobial activity is crucial for practical applications. This aspect is somewhat covered but could be expanded in the discussion section.
Missing Controls: the manuscript would be strengthened by including control experiments where only the base materials (e.g., AV and PVP without Thymol) are tested for antimicrobial activity. This would help isolate the effect of Thymol in the formulation.
Specific comments:
Figure 11 (Title/Caption Discrepancy): The title and caption of Figure 11 mention both Thymol and Thyme, but the actual content focuses only on Thymol. This should be corrected to avoid confusion (line 112).
Table 5 (Data Presentation): The presentation in Table 5 mixes several formulations without clear distinctions in concentrations and conditions, leading to potential confusion about the results' interpretation. The caption should explicitly detail these aspects to improve clarity (line 224-230).
Addressing general questions:
Clarity and structure: the manuscript is generally clear and well-structured. The introduction and background sections provide a robust framework for understanding the significance of the research.
Recentness and relevance of cited references: the references are mostly recent and relevant. However, it could benefit from more current literature to support the novelty of the research, especially concerning the mechanisms of Thymol's antimicrobial action.
Scientific soundness and experimental design: the experimental design is appropriate for testing the stated hypothesis. The methods are scientifically sound, though additional controls could enhance the robustness of the conclusions.
Reproducibility of results: the details provided are sufficient for reproducibility, although specifying concentrations in all tests would enhance this aspect.
Appropriateness of figures/tables: most figures and tables effectively support the data, but as noted, some could be clearer. The statistical analysis is well-handled, grounding the data interpretation solidly.
Consistency of conclusions: the conclusions are well-supported by the evidence provided. They align with the experimental data and discussion points throughout the manuscript.
Ethics and data statements: the manuscript includes appropriate ethics and data availability statements, ensuring compliance with journal standards.
See detailed specific comments-questions in the attachment.
Author Response
Dear Reviewer, thank you so much for your comments !
Reviewer 1
Article Title: Thymol and polymeric iodophor formulations: Investigating antimicrobial activities
Authors: Zehra Edis, Samir Haj Bloukh
Journal: International Journal of Molecular Sciences
Brief Summary: the study explores the effectiveness of Thymol within a polymeric iodophor formulation (AV-PVP-Thymol-I2) as an antimicrobial agent against various pathogenic microorganisms. The paper outlines the preparation and characterization of the formulation and assesses its inhibitory effects on selected reference strains through multiple analytical techniques, including SEM/EDS, UV-vis, Raman, FTIR, and XRD analyses. The primary strengths of the paper are its detailed experimental design and comprehensive analysis, which support the potential of AV-PVP-Thymol-I2 as a promising antimicrobial agent.
Dear Reviewer, thank you so much for your kind words!
General concept comments:
Areas of Weakness: the paper occasionally lacks specific details about the concentrations used in different formulations, particularly in Table 5's captions. This omission could lead to ambiguities in replicating the study or assessing the antimicrobial effectiveness accurately.
Dear Reviewer, you are right, thank you so much for this comment. We investigate iodine-PVP systems and kept on Table 5 only the concentrations of iodine. To avoid confusion, we moved the Thymol concentration towards the materials and Methods section, which step by step explains, what was done in this straight-forward-one-pot formulation. We hope, the questions are answered, when following the Materials and Methods section ““3.3. Preparation of AV-PVP-Thymol-I2”. For further comments please see below. Thanks
The distinction between the effects of Thymol and Thyme within the formulations is not clearly delineated, which could be misleading, as seen in Figure 11 where both are mentioned but only Thymol is shown.
Dear Reviewer, thank you so much for this comment. Unfortunately it was a typing mistake. I exchanged it with Thymol (line 709). Sorry for the confusion.
Testability of the Hypothesis: the hypothesis that Thymol incorporated in AV-PVP-I2 can enhance antimicrobial properties is well-framed and tested through rigorous methodologies. However, the study would benefit from additional controls, such as testing the formulation against a broader range of bacteria and fungi to validate the spectrum of activity.
Dear Reviewer, you are right, thank you so much for this important point. Actually, in our studies we depend on the standard, international collection microorganisms, a selection of 10 pathogens available in our research lab since several years. This is giving us a possibility to compare the results of new formulations against the same pathogens. Additionally, any other investigation using the same reference strain is also directly comparable to our studies. Of course, it is needed to get clinical microorganisms form hospital settings. At one stage we tried, faced a lot of but problems and did not succeed in applying this into our research lab, which is not part of any hospital, but instead is an academic institution. Maybe joint collaboration with existing, interested hospitals would be needed. This takes a lot of time, but we could try to find for our future research the right research hospitals for cooperation. Further reference strains, than our 10 pathogens are difficult to get due to international restrictions on the sale of microorganisms into different regions. Once lifted, we could try those as well. For now, unfortunately, we are left with our strains, consisting of C. albicans, 5 Gram-positive and 4 Gram-negative pathogens. I hope, you can understand our situation. Thank you so much.
Methodological Inaccuracies: the paper should address the stability of the formulation over time, as the long-term efficacy of the antimicrobial activity is crucial for practical applications. This aspect is somewhat covered but could be expanded in the discussion section.
Dear Reviewer, thank you so much. Following covers this in the discussion part of the manuscript:
Lines 834-846: “Storing the stock solution AV-PVP-Thymol-I2 for 25 months results in slightly reduced antimicrobial activities. In contrast, the Thyme-extract based AV-PVP-Thyme-I2 shows enhanced antimicrobial properties against the same 10 pathogens after 18 months [42]. Consequently, Thyme extract biocomponents in AV-PVP-Thyme-I2 like Thymol, Carvacrol and Rosmarinic acid are responsible together with AV components for sustaining the inhibitory action during its storage [42]. The title formulation AV-PVP-Thymol-I2 consists of Thymol alone in combination with AV biocomponents. During its storage up to 25 months, the polymeric iodophor PVP-I2 releases iodine due to the lack of key ingredients Carvacrol, Rosmarinic acid and others originating from the natural Thyme extract [42]. Thymol alone cannot sustain the sustained release reservoir. Additionally, Thymol has strong hydrogen bonding properties and over time replaces triiodide ions on the PVP-backbone, thus reducing slowly the inhibitory action of AV-PVP-Thymol-I2 (Figure 14) [79].”
Explanation: The stock solution was kept for 25 months in the fridge, but not the impregnated sutures, bandages, masks and disks. Coating those materials with the formulation and storing them would need airtight packaging under protective atmosphere. We recommend to use the stock solution for longer storage, while material applications may not be effective up to 25 months due to the volatile essential oil Thymol.
Missing Controls: the manuscript would be strengthened by including control experiments where only the base materials (e.g., AV and PVP without Thymol) are tested for antimicrobial activity. This would help isolate the effect of Thymol in the formulation.
Dear Reviewer, thank you so much for this comment. Yes, we did the needed antimicrobial tests. We changed table 5, separated some of the contents and added them into a new table 6. In this table, we added also the controls.
The antimicrobial results of AV alone and PVP alone are really very saddening compared to Thymol alone. Our previous publications also mention these AV and PVP antimicrobial activities [45,46]. If AV and PVP were alone effectively inhibiting, we would have never added Thymol ?. Actually, we were surprised with the AV results, because everywhere in the literature, AV is mentioned as antimicrobial plant. It has some inhibitory action, but its activity depends on soil, watering quality, environment, climate, location, type, watering procedure, temperature, harvesting time, harvesting season, harvesting and processing method. We achieved different results for summer and winter AV. We did not publish these results, because we like to write a long investigation about this subject. However, AV remains a miracle plant from nature and properly dealt with, it is a real miracle plant. We did a series of AV-PVP formulations with different solvents, harvesting time and different processing methods. We even tried with AV leave, rind or gel alone. Kindly understand, that we cannot share now the results of AV alone and PVP alone. We need some time to finalize our results. Anyhow, we changed the table 5 to reduce confusion, made another table 6 and added the recently performed control tests for PVP-I2 at the same concentration of 11µg/mL on discs as well as other data, which were removed from Table 5.
We added following sentences into the manuscript:
“We tested the control formulation PVP-I2 to judge the antimicrobial effectivity of our title compound AV-PVP-Thymol-I2 (Table 6). All 10 pathogens are more susceptible to the title compound with only one resistance against P. mirabilis ATCC 29906 (Table 6). In comparison, two strains, P. mirabilis ATCC 29906 and P. aeruginosa WDCM 00026, are resistant against PVP-I2 (Table 6). Inhibitory action of previously reported AV-PVP-I2 is much lower in comparison to AV-PVP-Thymol-I2 against the same selection of pathogens [46]. Adding AV enhances the antimicrobial properties of the formulation. These findings confirm the superiority of the title compound compared to the control PVP-I2 and AV-PVP-I2 (Table 6). As a result, adding Thymol and AV increased the susceptibility of the microorganisms against the title formulation (Table 6).” (lines 604-614)
Specific comments:
Figure 11 (Title/Caption Discrepancy): The title and caption of Figure 11 mention both Thymol and Thyme, but the actual content focuses only on Thymol. This should be corrected to avoid confusion (line 112).
Thank you so much. Corrected.
Table 5 (Data Presentation): The presentation in Table 5 mixes several formulations without clear distinctions in concentrations and conditions, leading to potential confusion about the results' interpretation. The caption should explicitly detail these aspects to improve clarity (line 224-230).
Dear Reviewer, thank you for this question. Actually, we never mentioned until today anything else than the concentrations of iodine in our work, because we are specialized in our research on iodine since long. Therefore, we also never change the iodine content throughout our investigations in order to be able to compare all our results over the years. As usual, we use on discs 11 µg/mL (1+), 2 mL of 5.5 µg/mL (2+) and 2 mL of 2.75 µg/mL (3+) of AV-PVP-Thymol-I2 (concentrations based on iodine content). On all the other materials (sutures, masks, bandage) we use only 11 µg/mL. To avoid confusion, we removed all unrelated data and put them together in Table 6.
Regarding further explanations, please see Materials and Methods section (lines 874-884):
“3.3. Preparation of AV-PVP-Thymol-I2
AV-PVP-Thymol-I2 is synthesized through a straightforward one-pot process. Firstly, 2 mL of pure AV gel is dispensed into a sterile beaker. Subsequently, 2 mL of a recently prepared solution containing 1 g of polyvinylpyrrolidone K-30 (PVP) dissolved in 10 mL of distilled water is introduced into the beaker under stirring at ambient temperature. Following this, 2 mL of Thymol solution (0.15 g in 10 mL of ethanol, 100 µg/mL) is slowly added to the mixture while stirring. Finally, 2 mL of a freshly prepared iodine solution (0.05 g of iodine in 3 mL of absolute ethanol) is incorporated into the mixture under continuous stirring and at room temperature. The resulting formulation, AV-PVP-Thymol-I2, is promptly transferred into a screw capped sterile glass sample tube and stored in darkness at 3°C in a refrigerator for subsequent use.”
Addressing general questions:
Clarity and structure: the manuscript is generally clear and well-structured. The introduction and background sections provide a robust framework for understanding the significance of the research.
Dear Reviewer, thank you so much.
Recentness and relevance of cited references: the references are mostly recent and relevant. However, it could benefit from more current literature to support the novelty of the research, especially concerning the mechanisms of Thymol's antimicrobial action.
Dear Reviewer, thank you for the question.
Our most recent references are
[79] Mangiacapre, E.; Triolo, A.; Ramondo, F.; Lo Celso, F.; Russina, O. Unveiling the structural organisation of carvacrol through X-ray scattering and molecular Dynamics: A comparative study with liquid thymol. J. Mol. Liquids 2024, 394, 123778. https://doi.org/10.1016/j.molliq.2023.123778
[71] Parolin, G.A.; Vital, V.G.; de Vasconcellos, S.P.; Lago, J.H.G.; Péres, L.O. Thymol as Starting Material for the Development of a Biobased Material with Enhanced Antimicrobial Activity: Synthesis, Characterization, and Potential Application. Molecules 2024, 29, 1010. https://doi.org/10.3390/molecules29051010
[73] Rojas, A.; Misic, D.; Dicastillo, C.L.; Zizovic, I.; Velásquez, E.; Gutiérrez, D.; Aguila, G.; Vidal, C.P.; Guarda, A.; Galotto, M.J. A review on thymol-based bioactive materials for food packaging. Ind. Crop Prod. 2023, 202, 116977.
[74] Hajibonabi, A.; Yekani, M.; Sharifi, S.; Nahad, J.S.; Dizaj, S.M.; Memar, M.Y. Antimicrobial activity of nanoformulations of carvacrol and thymol: New trend and applications. OpenNano 2023, 13, 100170.
[68] Zhao, A.; Zhang, Y.; Li, F.; Chen, L.; Huang, X. Analysis of the Antibacterial Properties of Compound Essential Oil and the Main Antibacterial Components of Unilateral Essential Oils. Molecules 2023, 28, 6304. https://doi.org/10.3390/molecules28176304
[82] Wijesundara, N.M.; Lee, S.F.; Cheng, Z.; Davidson, R.; Langelaan, D.N.; Rupasinghe, H.P.V. Bactericidal Activity of Carvacrol against Streptococcus pyogenes Involves Alteration of Membrane Fluidity and Integrity through Interaction with Membrane Phospholipids. Pharmaceutics 2022, 14, 1992. https://doi.org/10.3390/pharmaceutics14101992
[86] Wijesundara, N.M.; Lee, S.F.; Cheng, Z. Davidson, R.; Rupasinghe, H.P.V. Carvacrol exhibits rapid bactericidal activity against Streptococcus pyogenes through cell membrane damage. Sci Rep 2021, 11, 1487. https://doi.org/10.1038/s41598-020-79713-0.
These explain the antimicrobial actions of thymol in detail.
We added the following, previously missing information into the introduction (line 97-105):
“Thymol is a strong antimicrobial agent, which can easily enter cell membranes due to its small size and structural properties [68,71,73,79,81,82,86]. Thymol can enter like carvacrol through the microbial cell membranes [82,86]. Thymol can pass through the pores in the Gram-negative bacterial cell membrane [68,71,73,79,81,82,86]. The polar hydroxyl group interacts through hydrogen bonding with the polar cell membrane components and compromises cell functions [68,71,73,79,81,82,86]. Its nonpolar, aromatic ring engages with lipid bilayers and changes their fluidity and flexibility [82,86]. The outcome is rupture of the cell membrane, leaking of cell contents and finally bacterial death [68,71,73,79,81,82,86].”
(Similar action of Thymol was also explained in the discussion already: Zhou et al. studied pure Thymol at a concentration of 100 µg/mL against S. aureus strains and reported, that slight resistance phenomenon occurs after 30 generation passages with Thymol [78]. Thymol showed fatal activity at a concentration of 100 µg/mL against several strains of the multi-drug resistant pathogen S. aureus by increasing cell membrane permeability through depleting NADPH [78]. The authors also indicated that a concentration of 400 µg/mL did not result in resistance against Thymol. They suggested that these results are due to the higher concentration of 400 µg/mL and the multifacetted targets of Thymol within the pathogen [78]. A lower concentration of Thymol (100 µg/mL) within the synergetic formulation of a plant-based iodophor AV-PVP-Thymol-I2 could present a better strategy to combat bacterial resistance.)
By this way, we also explained the main mechanisms of Thymols antimicrobial action in detailed way in the introduction. Thank you so much !
Scientific soundness and experimental design: the experimental design is appropriate for testing the stated hypothesis. The methods are scientifically sound, though additional controls could enhance the robustness of the conclusions.
Dear Reviewer, additional controls were made. We made sure, that all needed information is collected. Meanwhile our group is also working on AV alone and AV-PVP systems as explained above and cannot release part of the study. Some hints are available in our previous publications under the reference numbers 45 and 46 as explained above.
In general, the controls are not as good as the title formulation itself and not comparable with Thymol.
Please stay tuned, there is hopefully more to come from us….. ?
Thank you for your understanding.
Reproducibility of results: the details provided are sufficient for reproducibility, although specifying concentrations in all tests would enhance this aspect.
Dear Reviewer, only the iodine the concentrations were left in Table 5, Thymol concentration was removed, and added into Materials and Methods. The description is easily reproducible now and not confusing any more. Thanks a lot.
Appropriateness of figures/tables: most figures and tables effectively support the data, but as noted, some could be clearer. The statistical analysis is well-handled, grounding the data interpretation solidly.
Dear Reviewer, thanks a lot. We changed the confusing parts and also changed the Table 5.
Consistency of conclusions: the conclusions are well-supported by the evidence provided. They align with the experimental data and discussion points throughout the manuscript.
Dear Reviewer, thank you so much for this kind comment.
Ethics and data statements: the manuscript includes appropriate ethics and data availability statements, ensuring compliance with journal standards.
Thank you so much !
See detailed specific comments-questions in the attachment.
Dear Reviewer, your comments really improved our manuscript by your sound, direct and straightforward comments. We appreciate your efforts and support by reviewing our paper.
We hope, that we were able to satisfy all your concerns.
Again, thank you so much !!!
Best regards
Zehra
18.4.2024

Reviewer 2 Report
Comments and Suggestions for Authors
I found this article seems to be interesting and can be accepted after major revision of below comments:
What is the purpose of the morphological and biological assessment?
How is thymol incorporated into polymeric iodophor formulations?
What are the antimicrobial activities of the polymeric iodophor formulations containing thymol?
How does the presence of thymol enhance the antimicrobial properties of the polymeric iodophor formulations?
What are the potential applications of thymol-containing polymeric iodophor formulations in antimicrobial treatments?
Also refer to the below article:
https://doi.org/10.1016/j.matchemphys.2022.125770
Comments on the Quality of English Language
I found this article seems to be interesting and can be accepted after major revision of below comments:
What is the purpose of the morphological and biological assessment?
How is thymol incorporated into polymeric iodophor formulations?
What are the antimicrobial activities of the polymeric iodophor formulations containing thymol?
How does the presence of thymol enhance the antimicrobial properties of the polymeric iodophor formulations?
What are the potential applications of thymol-containing polymeric iodophor formulations in antimicrobial treatments?
Also refer to the below article:
https://doi.org/10.1016/j.matchemphys.2022.125770
Author Response
Reviewer 2
I found this article seems to be interesting and can be accepted after major revision of below comments:
Dear Reviewer, thank you so much for this kind comment. We hope, that we can clarify all your concerns.
What is the purpose of the morphological and biological assessment?
Dear Reviewer, thank you so much. The biological assessment was done to verify, if our formulation can be used as antimicrobial agent. Especially now, when there is antimicrobial resistance (AMR) is a big problem for the world population and its future, new, natural alternatives need to be studied to have an idea, of what can help against AMR. Therefore, we studied our formulation and many other before with the intention to have a glimpse into the possibilities, nature and plants can offer. Until now, we studied Sage and Thyme in the same combination and found, that Thyme has very good results. Thymol is one of the major constituents of Thyme, therefore, we needed to find out, what will happen, if we used Thymol instead of the Thyme extract. The results were higher antimicrobial action, slightly shorter stability span due to the lack of synergistic mechanism originating actually from a natural plant extract. Therefore, the outcome presented is, very high antimicrobial activities for Thymol up to 25 months.
Second part of the question is about morphology. We need to see, if the samples are crystalline, amorphous, uniform and homogenous in order to judge, how they can be incorporated into applications. Again, the discs, sutures, masks and bandages were also analysed by SEM, so we can see, if the formulation is properly coating the material selected. As last, the antimicrobial activities will of course depend on the coating process and the morphology of the studied formulation on different materials.
How is thymol incorporated into polymeric iodophor formulations?
Dear Reviewer, Thymol is firstly incorporated into the polymeric PVP-backbone by hydrogen bonding towards the carbonyl-group of the PVP, which is also seen on the FTIR. Once iodine is added, Thymol is released slowly, while triiodide ions are hydrogen bonded to the carbonyl groups of the PVP. Anyhow, PVP and iodine are more matching due to their sizes and therefore are used as microbicides since long. It is a power duo. The released Thymol engages into hydrogen bonding with the other biocomponents originating from Aloe Vera. Over time, Thymol can again be hydrogen bonded to the PVP, due to iodine release.
What are the antimicrobial activities of the polymeric iodophor formulations containing thymol?
Dear Reviewer, thank you for this question. Polymeric iodophor formulations of our series (AV-PVP-I2, Reference No 46) without Thymol are not as strongly antimicrobial. Once Thymol is added, the antimicrobial inhibition is enhanced. Thymol is a strong antimicrobial and can inhibit 7 out of 10 of our pathogens. PVP-I2 alone inhibits 8, while our title formulation inhibits 9 of the reference strains with very good to intermediate results. More explanations are added as answer for the following question. Kindly see below.
How does the presence of thymol enhance the antimicrobial properties of the polymeric iodophor formulations?
Dear Reviewer, this is a very good question. Thank you so much ! According to our antimicrobial studies of Thymol alone as control, we have recorded strong susceptibility of our reference strains towards Thymol. Therefore, there is a synergism between having Thymol in the formulation together with a polymeric iodophor. Thymol has the ability to break through the microbial cell walls through interactions. It has the polar hydroxyl group, which allows hydrogen bonding towards the polar cell wall components, especially in the outer cell membrane. Once it enters the peptidoglycan layer, it depends on it nonpolar, aromatic ring and attaches by this way to the cell wall components. This breaches the cell wall, compromising all the structure, leads to leakage, disturbs the cell metabolism, protein synthesis and finally results in cell death [71, 78-81].
Also, we added following lines into the manuscript:
“Thymol is a strong antimicrobial agent, which can easily enter cell membranes due to its small size and structural properties [68,71,73,79,81,82,86]. Thymol can enter like carvacrol through the microbial cell membranes [82,86]. Thymol can pass through the pores in the Gram-negative bacterial cell membrane [68,71,73,79,81,82,86]. The polar hydroxyl group interacts through hydrogen bonding with the polar cell membrane components and compromises cell functions [68,71,73,79,81,82,86]. Its nonpolar, aromatic ring engages with lipid bilayers and changes their fluidity and flexibility [82,86]. The outcome is rupture of the cell membrane, leaking of cell contents and finally bacterial death [68,71,73,79,81,82,86].” (lines 97-105)
What are the potential applications of thymol-containing polymeric iodophor formulations in antimicrobial treatments?
Thymol containing polymeric iodophor formulations mitigate infective agents in enhanced way due to their synergy. Thymol alone is resistant against P. mirabilis, P. aeruginosa and S. pneumonia, while AV-PVP-I2 and PVP-I2 are resistant against P. mirabilis alone [46]. AV-PVP-Thymol-I2 does not inhibit P. mirabilis only, and in comparison to AV-PVP-I2 the ZOI are all much higher for the Thymol formulation. The potential applications are surface disinfectants, contact killing agents. The formulation can be used for prevention of surgical site infections, as well as wound infections, oral treatments and surgeries, because it has very good to intermediate antimicrobial action against 9 of our pathogens. Wound infections would be reduced, because Thymol itself is able to counteract the spread of microorganisms in the wound site as mentioned in the article below.
In addition, we wrote into the manuscript following lines:
“We tested the control formulation PVP-I2 to judge the antimicrobial effectivity of our title compound AV-PVP-Thymol-I2 (Table 6). All 10 pathogens are more susceptible to the title compound with only one resistance against P. mirabilis ATCC 29906 (Table 6). In comparison, two strains, P. mirabilis ATCC 29906 and P. aeruginosa WDCM 00026, are resistant against PVP-I2 (Table 6). Inhibitory action of previously reported AV-PVP-I2 is much lower in comparison to AV-PVP-Thymol-I2 against the same selection of pathogens [46]. Adding AV enhances the antimicrobial properties of the formulation. These findings confirm the superiority of the title compound compared to the control PVP-I2 and AV-PVP-I2 (Table 6). As a result, adding Thymol and AV increased the susceptibility of the microorganisms against the title formulation (Table 6).” (lines 604-614)
The title formulation could also be used in health care and public spaces as disinfective agent. The use on masks can help to mitigate the spread of air-droplet based infections, and upper respiratory tract infections. Masks could be re-used by spraying our formulation on it.
We hope, the explanations are satisfying. Thank you so much for your comment.
Also refer to the below article:
https://doi.org/10.1016/j.matchemphys.2022.125770
Dear Reviewer, thank you for the informative article mentioning about, how the wound environment shall be “An ideal wound dress should be able to prevent germs from entering the wound site and with microbial contamination at the wound, counteract and prevent the spread of infection” I added it as reference 126.
Thank you so much for all your efforts to improve our manuscript. We hope, that we were able to answer your questions. Including the above parts into the manuscript surely enhanced the paper.
Thank you again so much
Best regards
Zehra and Samir
18.4.2024

Reviewer 3 Report
Comments and Suggestions for Authors
The study presented in the manuscript focuses on the antimicrobial properties of a formulation consisting of Aloe Barbadensis Miller (AV), Thymol, iodine (I2), and polyvinylpyrrolidone (PVP). The work claims to offer an alternative solution to combat antimicrobial resistance (AMR), which is a pressing global health concern. The research is timely, and the approach of utilizing natural compounds in synergy with iodophors is innovative. However, there are several issues concerning the originality, methodology, data presentation, and interpretation that need to be addressed:
1. The manuscript states that the combination of natural compounds with iodophors is an innovative approach. However, the use of Thymol and other plant-based antimicrobials is well-documented, and the combination with iodine-based compounds is not unprecedented. Therefore, the innovation claimed might be overstated unless the authors can demonstrate a novel mechanism of action or significantly enhanced efficacy.
2. The methods used to synthesize and characterize the AV-PVP-Thymol-I2 formulation are standard and appear to be adequate. However, the manuscript lacks a detailed description of certain procedures, such as the methods used for testing against microbial strains. The dilution and application methods for different materials (discs, sutures, masks) should be more clearly described to ensure reproducibility.
3. The manuscript would benefit from a more structured presentation of data, particularly in the antimicrobial activity results. Tables and figures should be clearly labeled and discussed within the text to help the reader understand the findings.
4. The supplementary materials are referenced but not provided in the review copy. For a thorough assessment, these would need to be included.
5. The manuscript contains several typographical errors, inconsistencies in abbreviation usage, and unclear statements that could lead to confusion. For instance, the term 'sustained release reservoir' is used without proper introduction or explanation.
6. The statistical analysis section mentions one-way ANOVA but does not provide any actual data or p-values to substantiate the claims made in the text.
7. Some of the figures and tables are referenced incorrectly or are missing (e.g., Figures 1 to 5, Tables 1 to 4 are not included in the provided text).
8. The tables provided in the text (e.g., Table 5) are comprehensive but could be overwhelming; simplifying the presentation of the data could help readability.
9. The conclusions drawn from the data appear to be overstated in some cases. The manuscript claims high efficacy of the formulation against various pathogens, but comparative data against current standards of care are not adequately presented.
10. The literature review seems extensive, yet it does not clearly delineate how this study's findings differ significantly from existing research. The authors should highlight the gaps in current research that their work addresses.
11. While the antimicrobial potential of the formulation is discussed, the manuscript does not address the potential for clinical application. Data on cytotoxicity, in vivo efficacy, and potential side effects are essential to support the claims made.
Comments on the Quality of English LanguageModerate editing of English language required
Author Response
Reviewer 3
The study presented in the manuscript focuses on the antimicrobial properties of a formulation consisting of Aloe Barbadensis Miller (AV), Thymol, iodine (I2), and polyvinylpyrrolidone (PVP). The work claims to offer an alternative solution to combat antimicrobial resistance (AMR), which is a pressing global health concern. The research is timely, and the approach of utilizing natural compounds in synergy with iodophors is innovative. However, there are several issues concerning the originality, methodology, data presentation, and interpretation that need to be addressed:
Dear Reviewer, thank you for your kind comments above. We will do our best to resolve all the issues addressed in the below comments and hope to improve the manuscript by your diligent efforts.
- The manuscript states that the combination of natural compounds with iodophors is an innovative approach. However, the use of Thymol and other plant-based antimicrobials is well-documented, and the combination with iodine-based compounds is not unprecedented. Therefore, the innovation claimed might be overstated unless the authors can demonstrate a novel mechanism of action or significantly enhanced efficacy.
Dear Reviewer, thank you for this interesting comment.
Yes, indeed, the use of Thymol and other plant-based antimicrobials are extremely well documented in the literature. A growing number of investigations deals with the target to find solutions against AMR based on these natural ingredients. However, a smaller group of studies deal with combinations of polymeric iodophors and plant extracts. Our group is specialized in iodine-based antimicrobials. We studied until now different formulations of the iodophor PVP-I2 with Sage, Thyme, Pomegranate Peel and cinnamon. We also tested within this framework (same concentrations, same 10 pathogens, same methods) pure major components of the plants like trans-cinnamic acid and now thymol. Our aim is to categorize and find solutions against AMR by understanding the antimicrobial potential of plants, their components and their mixtures. It is a long journey.
A novel mechanism cannot be claimed, but our findings indicate better inhibitory actions of different combinations. We started our journey with simple formulations of PVP-I2, than increased the spectrum by adding plant extracts and finally essential oils [46]. While PVP-I2 shows in Table 6 resistance against 2 Gram-negative pathogens, adding AV to the formulation (AV-PVP-I2) slightly increases the ZOI and drops the resistant number of reference strains to one (P. mirabilis). Finally adding Thymol to AV-PVP-I2, which is our title formulation indeed, we see much higher antimicrobial action against all the pathogens used without change in the resistance of P. mirabilis.
In order to make it easier and less confusing, we restructured Table 5, removed data and created another Table 6. We also explained the new Table 6 accordingly in the manuscript:
“We tested the control formulation PVP-I2 to judge the antimicrobial effectivity of our title compound AV-PVP-Thymol-I2 (Table 6). All 10 pathogens are more susceptible to the title compound with only one resistance against P. mirabilis ATCC 29906 (Table 6). In comparison, two strains, P. mirabilis ATCC 29906 and P. aeruginosa WDCM 00026, are resistant against PVP-I2 (Table 6). Inhibitory action of previously reported AV-PVP-I2 is much lower in comparison to AV-PVP-Thymol-I2 against the same selection of pathogens [46]. Adding AV enhances the antimicrobial properties of the formulation. These findings confirm the superiority of the title compound compared to the control PVP-I2 and AV-PVP-I2 (Table 6). As a result, adding Thymol and AV increased the susceptibility of the microorganisms against the title formulation (Table 6).” (lines 604-614)
Thank you so much for improving our manuscript.
- The methods used to synthesize and characterize the AV-PVP-Thymol-I2 formulation are standard and appear to be adequate. However, the manuscript lacks a detailed description of certain procedures, such as the methods used for testing against microbial strains. The dilution and application methods for different materials (discs, sutures, masks) should be more clearly described to ensure reproducibility.
Dear Reviewer, thank you for this comment. As described above, our methods are used since long during our series of studies and publications by our experienced team members. We do not change much, so we can stay reproducible. All the methods are based on previous investigations to ensure also comparative results in order to investigate into better options [42-46]. The methods are the Kirby-Bauer methods [127] and we follow strictly the Clinical and Laboratory Standards Institute (CLSI) for conducting antimicrobial testing [128]. These methods and materials are explained in the part “Materials and Methods”.
We noticed, that there are few missing items and added them into that section:
1-We added the missing information, that the bacterial inoculum suspension was prepared with a concentration of 1 x 106 CFU/mL (OD600 = 0.02). (line 952).
2-We added the references, so the readers can take a look at our methods in our previous similar investigations [42-46].
3-Otherwise, as explained in the materials and methods sections, we follow the disc-dilution method by impregnating 2 ml of formulation with a stock concentration of 11 µg/mL on the materials. This was added in the line 993. The sizes of the material as well as the method of impregnation is described there. It is simple dip-coating.
- The manuscript would benefit from a more structured presentation of data, particularly in the antimicrobial activity results. Tables and figures should be clearly labeled and discussed within the text to help the reader understand the findings.
Dear Reviewer, we thank you for this comment. Indeed, we divided Table 5 into two tables. This will reduce the confusion, because table 5 was containing too much information and not ordered. Now, Table 5 is only the data related to the title formulation.
Table 6 is any other needed information like comparative data and controls. We also re-structured accordingly the flow of the text.
- The supplementary materials are referenced but not provided in the review copy. For a thorough assessment, these would need to be included.
Dear Reviewer, we submitted the supplementary materials. We do not know, why they were not provided. Maybe, now with this revision, you may be provided with the supplementary material. Surely, these will be published along the paper.
- The manuscript contains several typographical errors, inconsistencies in abbreviation usage, and unclear statements that could lead to confusion. For instance, the term 'sustained release reservoir' is used without proper introduction or explanation.
Dear Reviewer, we thoroughly read the manuscript to correct typos, abbreviation usage mistakes and unclear statements.
We also explained the term sustained release reservoir in lines 72-76 as follows:
“PVP attaches triiodide ions through hydrogen bonding and protects them from premature release. According to a study of Ma et al., PVP is termed as a sustained release reservoir for iodine [24]. Interaction and exposure of PVP with polar molecules, light or oxygen is detrimental for the antimicrobial properties because triiodide ions are released [24].”
- The statistical analysis section mentions one-way ANOVA but does not provide any actual data or p-values to substantiate the claims made in the text.
Dear Reviewer, thank you for this opportunity to explain our methods. As an experienced team on this type of investigation, we have already developed a protocol not mentioned in our paper. We do not do only triplicate, if there is a difference in our results indicating a deviation from previous results. If we find statistically significant differences (p above 0.05) between row-based values through Pearson correlation, we repeat the tests several times until we find the source of the problem. If there is no problem, we will continue until we really achieve no statistically significant differences (p > 0.05) between row-based values through Pearson correlation. This is expressed in very similar ZOI, measured in mm. Sometimes, we find out, we used by mistake expired products, or received from the factory slightly deviant products. Problems arise possibly like antimicrobial discs are no more giving the same control ZOI as usual, then discs are expired or faulty and we complain to the company to get refunded and proper products. Another issue sometimes arises in the used Agar plates. As you have noticed, we but already prepared Agar plates in order to reduce human error and deviation. Also here, it happens, that few times, some Agar plates are slightly different. Our team is dedicated to quality of products and ethical reporting. In short, if we see deviation, we repeat until we have no significant differences (p > 0.05) by finding the root causes and eliminating them. If we do not follow this practice, we cannot trust our own results and build on them. We can assure you, that the results are according to the claims.
- Some of the figures and tables are referenced incorrectly or are missing (e.g., Figures 1 to 5, Tables 1 to 4 are not included in the provided text).
Dear Reviewer, thank you for this comment. We checked and corrected mistakes in Figures 1 to 5 related descriptions etc. We did not find any missing figure or table in our text otherwise. Thank you so much for your comments.
- The tables provided in the text (e.g., Table 5) are comprehensive but could be overwhelming; simplifying the presentation of the data could help readability.
Thank you, dear Reviewer, we divided the table 5 into two as explained above. We believe, now it is easier to read the text. Your comment improved the manuscript extremely.
- The conclusions drawn from the data appear to be overstated in some cases. The manuscript claims high efficacy of the formulation against various pathogens, but comparative data against current standards of care are not adequately presented.
Thank you for your comment, dear Reviewer. We are sorry for the overstatements. Kindly see this as our excitement about these results towards our selection of 10 reference strains in comparison to our previous investigations [42-46].
Dear Reviewer, you are right, thank you so much for the important point regarding the comparative data against current standards of care. Actually, in our studies we depend on the standard, international collection microorganisms, a selection of 10 pathogens available in our research lab since several years. This is giving us a possibility to compare the results of new formulations against the same pathogens. Additionally, any other investigation using the same reference strain is also directly comparable to our studies. Of course, it is needed to get clinical microorganisms form hospital settings. At one stage we tried, faced a lot of but problems and did not succeed in applying this into our research lab, which is not part of any hospital, but instead is an academic institution. Maybe joint collaboration with existing, interested hospitals would be needed. This takes a lot of time, but we could try to find for our future research the right research hospitals for cooperation. Further reference strains, than our 10 pathogens are difficult to get due to international restrictions on the sale of microorganisms into different regions. Once lifted, we could try those as well. For now, unfortunately, we are left with our strains, consisting of C. albicans, 5 Gram-positive and 4 Gram-negative pathogens. I hope, you can understand our situation. Thank you so much.
- The literature review seems extensive, yet it does not clearly delineate how this study's findings differ significantly from existing research. The authors should highlight the gaps in current research that their work addresses.
Dear Reviewer, we started with this journey by adding AV to polymeric iodophors, which was innovative at that time. We then proceeded to add plant extract to this formulation. Of course, there are a lot of studies related to Thymol in the literature. They appear in combination with other essential oils, plant-extracts, nanoparticles, chitosan (added to line 96) as given in the references as expressed in the introduction briefly. A study from Sharma et al. [83] studied the combination of AV and chitosan encapsulated Thymol as an antimicrobial agent successfully (added to lines 107-108).” A study from Sharma et al. studied the combination of AV and chitosan encapsulated Thymol as an antimicrobial agent with very good inhibitory action [83].”
Regarding the investigation on PVP, there are only few cases of mixtures between povidone iodine and plant extracts, like the study of Rahma at al with curcumin [47] apart from our own studies (added to lines 80-81). “Rahma et al. studied PVP-curcumin combinations with good antimicrobial activities [47].”
Therefore, the gaps would be to encapsulate Thymol in PVP-I together with polyphenols and monoterpenes from plant extracts and combine them with essential oils. We believe, this will render good antimicrobial agents (added to lines 108-110). “Consequently, incorporating Thymol into PVP-I2 together with polyphenols or essential oils originating from plant extracts could render good antimicrobial agents.”
- While the antimicrobial potential of the formulation is discussed, the manuscript does not address the potential for clinical application. Data on cytotoxicity, in vivo efficacy, and potential side effects are essential to support the claims made.
Dear Reviewer, we agree with you point of view. Therefore, we acknowledged, that these studies still need to be done in the conclusions. This will require few more years to accomplish the full information. For our group, we do not have the infrastructure for such studies. We would need to outsource several tests and find the right partners for collaboration.
“Further research is necessary to assess its biological activities in vivo, cytotoxicity and potential side effects.” (added as last sentence of the manuscript).
Thank you for this suggestion and your supportive comments.
Dear Reviewer, thank you for your detailed study of our manuscript. Your efforts surely improved the readability and understanding of our manuscript, as well as the general structure and presented information. We hope, that we were able to address all your comments properly.
Thank you so much again.
Best regards
Zehra and Samir
18.4.2024

Round 2
Reviewer 2 Report
Comments and Suggestions for Authors
accept
Reviewer 3 Report
Comments and Suggestions for Authors
This manuscript is available for acceptance.
Comments on the Quality of English LanguageMinor editing of English language required